# Effect of changing vegetation and precipitation on denudation (part 2): Predicted landscape response to transient climate and vegetation cover over millennial to million year timescales

Manuel Schmid [1], Todd A. Ehlers[1]*, Christian Werner.[2] ,Thomas Hickler[2][3], Juan-Pablo Fuentes-Espoz[4]

[1] University of Tuebingen, Department of Geosciences; Wilhelmstrasse 56, 72074 Tuebingen, Germany (Manuel.Schmid@Uni-Tuebingen.de, Todd.Ehlers@Uni-Tuebingen.de)

[2] Senckenberg Biodiversity and Climate Research Center (BiK-F), Senckenberganlage 25, 60325 Frankfurt/Main, Germany (Christian.Werner@Senckenberg.de)

[3] Department of Physical Geography, Geosciences, Goethe-University, Frankfurt, Altenhoeferallee 1, 60438 Frankfurt/Main, Germany (Thomas.Hickler@Senckenberg.de)

[4] University of Chile, Department of Silviculture and Nature Conservation, Av. Santa Rosa 11315, La Pintana, Santiago RM, Chile (jufuente@uchile.cl)

* Corresponding author: (Todd.Ehlers@Uni-Tuebingen.de)

**Abstract** We present a numerical modeling investigation into the interactions between transient climate and vegetation cover with hillslope and detachment limited fluvial processes. Model simulations were designed to investigate topographic patterns and behavior resulting from changing climate and associated changes in surface vegetation cover. The Landlab surface process model was modified to evaluate the effects of temporal variations in vegetation cover on hillslope diffusion and fluvial erosion. A suite of simulations were conducted to represent present day climatic conditions and satellite derived vegetation cover at four different research areas in the Chilean Coastal Cordillera. These simulations included steady-state simulations as well as transient simulations with forcings in either climate or vegetation cover over millennial to million-year timescales. These simulations included two different transient variations in climate and vegetation cover including a step change in climate or vegetation, as well as 100 kyr oscillations over 5 Myr. We conducted eight different step-change simulations for positive and negative perturbations in either vegetation cover or climate and six simulations with oscillating transient forcings for either vegetation cover or climate, and oscillations in both vegetation cover and climate. Results indicate that the coupled influence of surface vegetation cover and mean annual precipitation shifts basin landforms towards a new steady state, with the magnitude of change highly sensitive to the initial vegetation and climate conditions of the basin. Dry, non-vegetated basins show higher magnitudes of adjustment than basins that are situated in wetter conditions with higher vegetation cover. For coupled conditions when surface vegetation cover and mean annual precipitation change simultaneously, the landscape response tends to be weaker. When vegetation cover and mean annual precipitation change independently from each other, higher magnitude shifts in topographic metrics are predicted. Changes in vegetation cover show a higher impact on topography for low initial surface cover values whereas for areas with high initial surface cover, the effect of changes in precipitation dominate the formation of landscapes. This study demonstrates a sensitivity of catchment characteristics to different transient forcings in vegetation cover and mean annual precipitation, with a crucial role for initial vegetation and climate conditions.

## 1. Introduction

Plants cover most of Earth's surface and interact chemically and physically with the atmosphere, lithosphere and hydrosphere. The abundance and distribution of plants throughout Earth's history is a function, amongst other things, of changing climate conditions that can impact the temporal distribution of plant functional types and vegetation cover present in an area (Hughes, 2000; Muhs et al., 2001; Walther et al. 2002). The physical feedbacks of vegetation on the Earth's near surface manifest themselves mainly through an influence of plants on weathering, erosion, transport and the deposition of sediments (Marston, 2010; Amundson et al., 2015). Although the effects of biota on surface processes has been recognized for over a 100 years (e.g., Gilbert, 1877; Langbein & Schumm, 1958), early studies focused mainly on qualitative descriptions of the underlying processes. With the rise of new techniques to quantify mass transport from the plot- to catchment-scale, and the emergence of improved computing techniques and landscape evolution models, research shifted more towards building a quantitative understanding of how biota influence both hillslope and fluvial processes (Stephan and Gutknecht, 2002; Roering et al., 2002; Marston, 2010; Curran and Hession, 2013). The previous studies motivate the companion papers presented here. In part 1 (Werner et al. 2018-this volume) a dynamic vegetation model is used to evaluate the magnitude of past (Last Glacial Maximum to present) vegetation change along the climate and ecological gradient in the Coastal Cordillera of Chile. Part 2 (this study) presents a sensitivity analysis of how transient climate and vegetation impact catchment denudation. This component is evaluated through implementation of transient vegetation effects for hillslopes and detachment limited rivers in a landscape evolution model.

Previous research in agricultural engineering has focused on plot-scale models to predict total soil loss in response to land-use change (Zhou et al. 2006; Feng et al. 2010) or general changes in plant surface cover (Gyssels et al. 2005), but do not draw conclusions about large-scale geomorphic feedbacks active over longer (millennial) timescales and larger spatial scales. However, a better understanding of how vegetation influences the large scale topographic features (e.g. relief, hillslope angles, catchment denudation) is crucial to understanding the evolution of modern landscapes. At the catchment scale, observational studies have found a correlation between higher values of mean vegetation cover and basin wide denudation rates or topographic metrics (Jeffery et al., 2014, Sangireddy et al. 2016, Acosta et al. 2015). Parallel to the previous observational studies, numerical modeling experiments of the interactions between landscape erosion and surface vegetation cover have also made progress. For example, Collins et al. (2004) were one of the first who attempted to couple vegetation dynamics with a landscape evolution model and found that the introduction of plants to their model resulted in steeper equilibrium landscapes with a higher variability in magnitude of erosional events. Following this, subsequent modeling studies built upon the previous findings with more sophisticated formulations of vegetation-erosion interactions (Istanbulluoglu and Bras, 2005) including the influence of root strength on hillslopes (Vergani et al., 2017). These studies found that not only is there a positive relationship between vegetation cover and mean catchment slope and elevation but there also exists an inverse relationship between vegetation cover and drainage density, due to the plants ability to hinder fluvial erosion and channel initiation.

The advances of the previous studies are limited mainly by their consideration of static vegetation cover or very simple formulations of dynamic vegetation cover. The exception to this is Istanbulluoglu and Bras (2005) who also considered the lag time for vegetation regrowth on hillslopes after a mass wasting event and Yetemen et al. (2015) which considered more complex hydrology in their models but on a smaller spatial scale. However, numerous studies (Ledru et al., 1997; Allen and Breshears, 1998; Bachelet et al., 2003) document that vegetation cover changes in tandem with climate change over a range of timescales (decadal to million year). Missing from previous landscape evolution studies, is consideration of not only how transient vegetation cover influences catchment denudation, but also how coeval changes in precipitation

influence catchment-wide mean denudation. While the effects of climate change over geologic timescales on denudation rates and sediment transport dynamics have been investigated by others (e.g., Schaller et al., 2002; Dosetto et al., 2010; McPhillips et al. 2013), the combined effects of vegetation and climate change on catchment denudation have not. Thus, over longer (geologic) timescales, we are left with a complicated situation of both vegetation and climate changes, and the individual contributions of these changes to catchment scale denudation are difficult to disentangle.

In this study, we build upon previous works by investigating both the temporal and spatial sensitivities of landscapes to the coupled vegetation-climate system. By focusing on simplified transient forcings such as a step change, or 100 kyr oscillations in climate and vegetation cover we present a sensitivity analysis of the landscape response to each of these changes, including a better understanding of the direction, magnitude and rates of landscape change. Our model setup is motivated by four study-areas along the climate and vegetation gradient in Chile (Fig. 1a) and illuminates the transient catchment response to biotic vs. climate changes. These study areas are part of the recently initiated German priority research program *EarthShape: Earth surface shaping by biota* (www.earthshape.net). This region is used to provide a basis for our model setup for covariation in precipitation and vegetation present in a natural setting. While we present results representative of the Coastal Cordillera, Chile, it is beyond the scope of this study to provide a detailed calibration to this area and our main objective is identifying the sensitivity, and emergent behaviour, of catchment denudation to changing precipitation and vegetation cover over millennial timescales. This study also builds upon results from the companion paper (Werner et al. 2018 - this volume) by imposing temporal variations in vegetation cover, identified in that study.

## 2. Background to model setup

Model setup and the range of initial conditions chosen for models were based upon four study-areas located in the Coastal Cordillera of Chile (26ºS to 38ºS). The focus areas shown in Fig. 1a were chosen because of their similar granitic lithology and geologic and tectonic history (Andriessen and Reutter, 1994; McInnes et al., 1999; Juez-Larré et al., 2010; Maksaev and Zentilli, 1999; Avdievitch et al., 2017), and the large gradient in climate and vegetation cover over the region (Fig.1b,c). These study areas include (from north to south): Parque Nacional Pan de Azúcar; Reserva Santa Gracia; Parque Nacional La Campana, and Parque Nacional Nahuelbuta. Although this study does not explicitly present landscape evolution model results 'calibrated' to these specific areas, we've chosen the model input (e.g. precipitation, initial vegetation cover, rate of tectonic rock uplift) to represent these areas to provide simulation results that represent the non-linear relationship between precipitation and vegetation cover (e.g. Fig. 1b, c) over a large climate gradient.

Topographic metrics such as mean basin slope, total basin relief, mean basin channel steepness, and mean surface vegetation cover and mean annual precipitation were extracted for the main catchments and a subset of adjacent catchments (Fig. 1; Fig. 2). Topographic metrics were extracted from 30 m resolution digital elevation model from the NASA shuttle radar topography mission (SRTM), and vegetation related datasets from the moderate resolution imaging spectroradiometer (MODIS) satellite data (https://landcover.usgs.gov/green_veg.php).

## 2.1 Landscape evolution modeling approach and the applicability of these results

Landscape evolution model studies can be assigned to different general approaches, which were conceptually defined by Dietrich et al. (2013). The different approaches presented in the Dietrich et al. (2013) study mostly differ in the complexity

of input parameters and in the resulting claim for reproducing realistic complexity in modelled landscapes. For this study
we have chosen the approach of essential realism, which acknowledges a system-inherent indeterminacy in the evolving
topography but focuses on predicting the first-order trends within a system and the differences between landscapes, based
on different external conditions, incorporated in the model (Howard, 1997).
While we do not claim to reproduce the topographic metrics of the four different focus areas in Chile on a realistic level,
our approach determines the general first-order effects of millennial timescale changes in precipitation and vegetation
cover that can impact topography. Superimposed on the effects documented in this study would be the effects of seasonal
changes in precipitation and vegetation cover, subcatchment variations in vegetation cover, transport limited fluvial and
vegetation interactions, stochastic variations in precipitation in different climate zones. Consider of the previous, more
detailed, aspects of precipitation-vegetation interactions on erosion could be independent studies of their own and can not
be covered in a single study. Thus, the modeling approach and results of this study should be considered as documenting
the longer (millennial) timescale climate and vegetation forcings on fluvial and surface processes.
**3. Methods**
**3.1. Model Description and governing equations**
For this study, we use the open-source model framework Landlab (Hobley et al., 2017). We chose a model-domain with
an area of 100 km$^2$ which is implemented as a rectangular grid, divided into 0.01km$^2$ spaced grid cells. For simplification
in the presentation of results, we present our results for the driest, northern most area (Parque Nacional Pan de Azucar)
and for the Parque Nacional La Campana. La Campana is situated at 32°S Latitude (Fig.1) and shows the highest values
in analyzed basin metrics (Fig. 2), although the general behavior and results presented here are representative of the other
two areas not shown. The topographic evolution of the landscape is a result of tectonic uplift and surface processes,
incorporating detachment limited fluvial erosion and linear diffusive transport of sediment across hillslopes (Fig.3). These
processes are linked to, and vary in their effectiveness due to surface vegetation density. Details of the implementation of
these processes into Landlab are explained in the following subsections.
This model setup is simplified in regards to hydrological parameters such as soil moisture and groundwater and
unsaturated zone flow. Also, the erosion and transport of material due to mass-wasting processes such as rockfalls and
landslides are not considered. We argue that those processes do not play a major role in the basins we used for model-
calibration and that the processes acting continuously along hillslopes and channels have the largest impact on shaping
our reference landscapes. The detachment-limited approach was chosen because the focus areas represent small, bedrock
dominated headwater catchments. Additional caveats and limitations of the modeling approach used are discussed in
Section 5.4. Main model parameters used in the model (and described below) are provided in Table 1.
**3.2. Boundary and Initial Conditions, and Model Free Parameters**
In an effort to keep simulations comparable, we minimized the differences in parameters between simulations. The
exceptions to this include the surface vegetation cover and mean annual precipitation, which were varied between
simulations. One of the main controls on topography is the rock uplift rate. We kept the rock uplift rate temporally and
spatially uniform across the domain and at 0.2 mm/yr (Table 1). Studies of the exhumation and rock uplift history of the
Coastal Cordillera, Chile, are sparse at the latitudes investigated here, but existing and in progress studies further to the
north are broadly consistent with the rock uplift rate used here (Juez-Larré et al., 2010; Avdievitch et al., 2017).
The EarthShape focus sites are situated in similar granitic lithologies (Oeser et al., 2018), thereby allowing the assumption
that the same critical shear-stress, baseline diffusivity, and fluvial erodibility can be used.
Vegetation cover was chosen to be spatially uniform across model domains. While vegetation can change in high-relief
catchments due to precipitation and temperature changes with elevation, this simplifying assumption was made based on
the low to moderate relief (500-1500m, mean ~750m) of the Coastal Cordillera areas investigated, and minimal field and
MODIS observed changes in type and cover with elevation.  The exception to this La Campana study area (~1,500 m
relief) which has an observed change in vegetation type and cover in the upper 500 m of the catchment.  Furthermore,
dynamic vegetation modeling results presented in the companion paper to this (Fig. 5b in Werner et al., 2018 - this
volume) indicate that although elevation gradients in plant functional types occur in the region since the last glacial
maximum, the elevation range of the catchments in the Coastal Cordillera (<1500 m) exhibits only minor changes with
elevation. Vegetation cover near trunk streams within catchments is observed in the field to increase, most likely due to
local scale hydrology and more abundant water in these areas.  However, these regions are often restricted to with 10's
of meters of the trunk stream, well below the 100m grid resolution of the model, and therefore difficult to accurately
resolve within the simulations presented.

The initial topography used in our simulations was a random white-noise topography with <1 m relief. To avoid unwanted
transients related to the formation of this initial topography we conduct simulations to produce an equilibrium topography
for each set of the different climate and vegetation scenarios (see below). These equilibrium topographies were produced
by running the model for 15 Myr until a topographic steady-state is reached. The equilibrium topography after 15 Myr
was used as the input topography for subsequent experiments that impose transient forcings in climate, vegetation, or
both (Fig. 4). The model simulation time shown in subsequent plots is the time since completion of this initial 15 Myr
steady-state topography development. In the results section, we present these results starting with differences in the initial
steady-state topographies (prior to imposing transient forcings) and  then add different levels of complexity by imposing
either: (1) a single transient step-change for the vegetation cover (Fig. 4b); (2) a step change in the mean annual
precipitation (Fig. 4d); (3) 100 kyr oscillations in the vegetation cover (Fig. 4a); (4) 100 kyr oscillations in the mean
annual precipitation (Fig. 4c); or (5) 100 kyr oscillation in both the vegetation cover and mean annual precipitation (both
Fig. 4a and 4c). This approach was used to produce a stepwise increase in model complexity for evaluating the individual,
and then combined, effects of fluvial and hillslope processes to different forcings.
The magnitude of induced rainfall transient forcings where based upon the present-day conditions along the Coastal
Cordillera study areas (Fig. 1b, c). The step change and oscillations in vegetation cover and mean annual precipitation
imposed on the experiments were designed to investigate vegetation and precipitation change effects on topography over
the last ~0.9 Ma, the period during which a 100 kyr orbital forcing is dominant in Earth's climate (Broecker & van Donk,
1970; Muller and MacDonald, 1997). Given this timescale of interest, we impose a 10% magnitude change in the step-
increase or decrease, or the amplitude change in oscillations for the vegetation cover. This magnitude of vegetation cover
change is supported by dynamic vegetation modeling of vegetation changes over glacial-interglacial cycles in Chile (see
companion paper by Werner et al. 2018-this journal) and to some degree elsewhere in the world (Allen et al, 2010, Prentice
et al. 2011, Huntley et al. 2013), however for the sake of simplicity we use a fixed forcing of +/-10% for all simulations
and not a spatially variable forcing which would be dependent on ecosystem behaviour for each separate area. We assume
that the present-day conditions of combined vegetation cover and mean annual precipitation along the north-south gradient
of the coastal cordillera are directly linked (Fig. 1b, c), and therefore follow an empirical approach based on the present
day mean annual precipitation which directly links to the present-day vegetation cover in Chile (Fig. 5). We do this by
associating each 10% change in vegetation cover (dV) with a corresponding change of mean annual precipitation (dP,
Fig. 5) present in the study areas considered. We impose a predefined, fixed amplitude, change in surface vegetation cover
as a transient forcing for simulations. For our prescribed changes in vegetation cover we then choose corresponding values
of mean annual precipitation based on the relationship shown in figure 1 and 5. The simulations were parameterized in
terms of changes in vegetation cover (instead of precipitation) for two reasons. First, the emphasis of this study is on
advancing our knowledge of how vegetation changes impact surface processes. Given this, we wanted to present results
based on reasonable changes in vegetation cover change. Second, results from the companion paper to this one (Werner
et al., 2018) suggest the Chilean Coastal Cordillera experiences +/- 10% changes in vegetation covers over the last 21
kyr. We adopt this result in this study as the current best estimate for the changes in the study areas considered. Thus, the
changes in precipitation and vegetation imposed in this study are empirically based on observations from the climate and
ecological gradient in the Coastal Cordillera.
The boundary conditions used in the model were the same for all simulations explained above (Fig. 3). One boundary
was held at a fixed elevation and open to flow outside the domain. The other three were allowed to increase in elevation
and had a zero-flux condition. This design for boundary conditions is similar to previous landscape evolution modeling
studies (Istanbulluoglu and Bras, 2005) and provides a means for analyzing the effects of different vegetation cover and
precipitation forcings on the individual catchment and subcatchment scale.
**3.3. Vegetation Cover Dependent Geomorphic Transport Laws**
The governing equation used for simulating topographic change in our experiments follows the continuity of mass.
Changes in elevation at different points of the model domain over time dz(x,y,t) depend on
$$\frac{\delta z(x,y)}{\delta t} = U - \frac{\delta z}{\delta t}|_{hillslope} - \frac{\delta z}{\delta t}|_{fluvial} \tag{1}$$
where z is elevation, x, y are lateral distance, t is time, U is the rock uplift rate, $\frac{\delta z}{\delta t}|_{hillslope}$ is the change in elevation due
to hillslope processes, $\frac{\delta z}{\delta t}|_{fluvial}$ is the change in elevation due to fluvial processes (Tucker et al., 2001a).
**3.3.1 Vegetation Cover Influenced Diffusive Hillslope Transport**
The change in topography in a landscape over time caused by hillslope-dependent diffusion can be characterized as:
$$\frac{\delta z}{\delta t}|_{hillslope} = -\nabla q_{sd} \tag{2}$$
Landscape evolution models characterize the flux of sediment $q_s$ either as a linear or non-linear function of surface slope
S (Culling, 1960; Fernandez and Dietrich, 1997). In order to keep the number of free parameters for the simulation to a
minimum, we used the linear description of hillslope diffusion:
$$q_{sd} = K_d S \tag{3}$$
Following the approach of (; Alberts et al., 1995; Dunne, 1996; Istanbulluoglu and Bras, 2005; Dunne et al., 2010), we
assign α the linear diffusion coefficient $K_d$ as a function of surface vegetation density V, an exponential coefficient α, and
a baseline diffusivity $K_b$, such that:
$$K_d = K_b e^{-(\alpha V)} \tag{4}$$

### 3.3.2 Vegetation Cover Influence on Overland Flow and Fluvial Erosion

Fluvial detachment-limited erosion of material due to water is calculated in this study by the widely-used stream-power-equation (Howard and Kerby, 1983; Howard et al., 1994; Whipple and Tucker, 1999; Braun and Willet, 2013):

$$\frac{\delta z}{\delta t}\Big|_{fluvial} = k_e(\tau - \tau_c)^p \ for \ \tau > \tau_c \tag{5}$$

In this equation $k_e$ represents the erodibility of the bed, $\tau$ is the bed shear stress which acts on the surface at each node, $\tau_c$ is the critical shear stress which needs to be overcome to erode the bed-material and p is a constant.

By following the approach of Istanbulluoglu and Bras (2005) and Istanbulluoglu et al. (2004), we reformulate the standard equation of shear-stress $\tau_b = \rho_w gRS$, where $\rho_w$ is the density of water, g is the acceleration of gravity, R is the hydraulic radius and S is the local slope, to a form which incorporates Manning's roughness to quantify the effect of vegetation cover on bed shear stress (Willgoose et al. , 1991, Istanbulluoglu et al., 2004):

$$\tau_v = \rho_w g(n_s + n_v)^{\frac{6}{10}} q^m S^n F_t \tag{6}$$

Here $n_s$ and $n_v$ represent Manning's numbers for bare soil and vegetated ground, q is the water-discharge per node which is approximated with the steady-state uniform precipitation per timestep P and the surface area per node A (q = A* P) and S is the local slope per node, m and n are constants. $n_v$ for each node is calculated as a function of the local surface vegetation cover

$$n_v = n_{vr}(\frac{V}{V_r})^w \tag{7}$$

with $n_{vr}$ being the Manning's number for a defined reference vegetation cover, V and $V_r$ being the vegetation cover at each node and the reference vegetation cover and w is an empirical scaling parameter.

The last variable in equation 6 represents the shear-stress partitioning ratio $F_T$ (after Foster 1982; Istanbulluoglu and Bras, 2005), which is used to scale the shear-stress at each node to the vegetation-cover present.

$$F_t = (\frac{n_s}{n_s + n_v})^{3/2} \tag{8}$$

By combining the formulation for shear stress out of equation 6 with the general stream-power equation 5 we formulate a new factor $K_v$ which represent the bed erodibility per node as a function of surface vegetation cover, which leads to a new expression of fluvial erosion

$$K_v = k_e \rho_w g(n_s + n_v)^{\frac{6}{10}} F_t \tag{9}$$

$$\frac{\delta z}{\delta t}\Big|_{fluvial} = K_v \ q^m S^n \tag{10}$$

### 3.4 Model Evaluation

Model performance was evaluated using the above equations and different initial vegetation covers and mean annual precipitation. Our focus in this study is on the general surface process response to different transient vegetation and climate conditions. Given this, topographic metrics of relief, mean slope, and normalized steepness index ($K_{sn}$) were computed from the model results and compared to observed values from the 30 m SRTM DEM for each of the four areas (Fig. 2). This was done to evaluate if our implementation of the governing equations in Section 3.4 produced topographies within reason of present day topographies in the four Chilean areas.  A more detailed model calibration is beyond the scope of this study, and not meaningful without additional observational constraints on key parameters such latitudinal variations in the rock uplift rate and erosivity. Our aim is not to reproduce the present day topography of the Coastal

Cordillera study areas but rather identify the sensitivity and emergent behaviour of vegetation-dependent surface
processes gradient of vegetation cover and precipitation in Chile.
**4. Results**
Our presentation of results is structured around three groups of simulations. These include: 1. steady-state simulations
where equilibrium topographies are calculated for different magnitudes of vegetation cover and identical precipitation
forcing. A second set of steady-state simulations with the same magnitudes vegetation cover as 1. but with different
precipitation forcings corresponding to each vegetation cover (Fig. 5, Section 4.1). 2. Simulations with a transient step-
change in either surface vegetation density or precipitation (Section 4.2) that is initiated after the landscape has reached
steady state. and 3. simulations with a transient 100 kyr oscillating time series of changing vegetation or precipitation that
occurs after the landscape has reached steady state (Section 4.3). For each group of transient simulations, we show the
topographic evolution with help of standard topographic metrics and the corresponding erosion rates after the induced
change.
**4.1 Equilibrium Topographic Metrics**
Topographic metrics from each of the four Chilean focus areas (Fig. 1a) were extracted for comparison to equilibrium
topographies predicted after 15 Myr of model simulation time. This comparison was done to document the model response
to changing vegetation cover (with climate held constant) and changing vegetation cover and precipitation, and also to
demonstrate the modeling approach employed throughout the rest of this study captures the general characteristics of
different topographic metrics along the Chilean Coastal Cordillera.
Analysis of the digital elevation model for each of our four Chilean focus areas illustrates observed changes in catchment
relief, slope, and channel steepness ($K_{sn}$) in relation to the surface vegetation (Fig. 7, red points) and latitude (Fig. 2). The
general trend in the observed metrics shows a non-linear increase in each metric until a maximum is reached for regions
with 70% vegetation cover. Following this, all observed metrics show a decline towards the area with 100% vegetation
cover.
The model predicted equilibrium topographies (Fig. 7a,b,c) from four different steady-state simulations with different
vegetation cover in each simulation and a constant mean annual precipitation (900 mm/yr) show a nearly linear increase
in all observed basin metrics with increasing vegetation cover and therefore do not reflect the overall trend observed from
the study areas (red line/symbols). For example, basin relief and slope are both under predicted for simulations with V <
100% (Fig.7a,b), and only the predicted maximum relief for a fully-vegetated simulation resembles the DEM maximum
value. For the normalized channel steepness, only two observed mean values (for V = 10% and 70%) lie within the range
of mean to maximum predicted $K_{sn}$ values (Fig.7c).
The resulting equilibrium topographies from simulations with different mean annual precipitation and vegetation cover
in each simulation (Fig.7d,e,f) show an improved representation of the general trend of the DEM data. The vegetation
cover and precipitation values used in these simulations come from the Chilean study areas (Fig. 1b, c; Fig. 5). In these
simulations, the maximum in the observed basin metrics is situated at values of V = 30% with a following slight decrease
in the metric for V = 30% to V = 70%, followed by a steeper decrease in metrics from V = 70% to V = 100%. Generally
the model-based results tend to underestimate the basin relief and overestimate the basin channel steepness (Fig.7d,f).
Variations in basin slope are captured for all but the non-vegetated state (Fig.7e).
Although the above comparison between the models and observations demonstrates a range of misfits between the two,
there are several key points worth noting. First, the model results shown are simplified in their setup (e.g. assuming
similar rock uplift rate, identical lithology and constants), and assume the present day topography is in steady state for
the comparison. Second, despite the previous simplifying assumptions, the degree of misfit between the observations and
model are surprisingly small when both variable vegetation and variable precipitation, are considered (Fig. 7d,e,f). Finally
(third), the general 'humped' shape curve observed in the Chilean areas is captured in the model predictions (Fig. 7d,e,f),
with the notable exception that the maximum in observed values occurs at a higher vegetation cover (V = 70%) than the
model predictions (V = 30%). Explanations for the possible source of these differences are revisited in the discussion
section.

**4.2 Transient Topography From a Step Change in Vegetation or Precipitation**

The evolution of topographic metrics after a induced instantaneous disturbance (Fig. 4) of either only the surface
vegetation cover (Fig. 8, green lines) or only the mean annual precipitation (Fig. 8, blue lines) is analyzed for changes in
topographic metrics for either a positive disturbance (Fig. 8a,b,c) or a negative disturbance (Fig.8d,e,f). This scenario was
chosen to analyze and isolate the effects of these specific transient forcings, and are useful for understanding more
complex changes in vegetation and precipitation presented later. Mean catchment erosion rates are also analyzed for their
evolution after the disturbance (Fig.9). For simplicity in presentation, results are shown for only two of the four Chilean
study areas with initial vegetation (V) and precipitation (P) values for vegetation covers of 10 and 70%, and precipitation
rates that correspond to these vegetation covers (i.e. P(V=10%) or P(V=70%)) (Fig. 5). The results described below show
a general positive correlation between all observed topographic metrics and surface vegetation cover and a negative
correlation between observed topographic metrics and mean annual precipitation.

**4.2.1 Positive Step Change in Vegetation Cover or Precipitation**

**Topographic Analysis**

A positive step change in vegetation cover (V) from V = 10% to V = 20% (solid green line Fig. 8a,b,c) leads to a factor
of 1.9, 1.42, and 2.1 change in mean basin relief (from 270 m to 520 m), mean basin slope (11.2° to 15.9°), and mean
basin channel steepness (108 $m^{-0.9}$ to 222 $m^{-0.9}$), respectively. The adjustment time until a new steady state in each metric
is reached is 3.1 Ma. The corresponding positive change in mean annual precipitation (solid blue lines, Fig. 8a,b,c) leads
to a decrease of mean basin relief to 176 m, mean basin slope to 8.6° and mean basin channel steepness to 67 $m^{-0.9}$. This
corresponds to a decrease by factors of 1.5, 1.2 and 1.6, respectively. The adjustment time to new steady state conditions
in this case are shorter and 1.1Ma (Fig.8a,b,c). A second feature of these results is the brief increase and then decrease in
basin average slope angles following the step change (Fig. 8b).
For simulations with V = 70% initial surface vegetation cover, a positive increase to V = 80% leads to an increase of
mean basin relief from 418 m to 474 m, mean basin slope from 15.5° to 16.8° and mean basin channel steepness from
172 $m^{-0.9}$ to 199 $m^{-0.9}$. This causes an increase in each metric by factors of 1.1, 1.1 and 1.2, respectively. The adjustment
time to steady-state conditions is 1.9Ma (dotted green lines, Fig 8a,b,c). The corresponding positive change in mean
annual precipitation leads to a decrease of relief to 268 m, decrease in slope to 11.9° and decrease of channel steepness
to 105 $m^{-0.9}$. This resembles a decrease by factors 1.5, 1.3, 1.6, respectively. Adjustment time in this case is 1.7Ma (dotted
blue lines, Fig. 8a,b,c). The basin slope data shows similar behavior as the $V_{ini}$ = 10% simulations with an initial decrease
and then increase for a vegetation cover step change and an initial increase and then decrease for a step change in mean
annual precipitation. Comparison of the change in the topographic metrics for the low (V=10%) and high (V=70%) initial
vegetation covers, the magnitude of change in each metric is larger when the step change occurs on a low, rather than
higher, initial vegetation cover topography.
**Erosion Rate Changes**
The model results show a negative relationship between increases in vegetation cover and erosion and a positive
relationship between increases in precipitation and erosion (Fig. 9). Although the response between the disturbances and
changes in erosion rates are instantaneous, the maximum or minimum in the change is reached after some lag time and
the magnitude and duration of non-equilibrium erosion rates varies between different simulation setups.
For initial vegetation cover of V = 10%, a change in vegetation cover (dV) of +10% leads to a decrease in erosion rates
from 0.2 to 0.03 mm/yr (factor of 5.7 decrease, Fig. 9a green line). The minimum erosion rate is reached 43.5 kyrs after
the step change occurs. Following this minimum in erosion rates, the rates increase until the steady-state erosion rate is
reached after the adjustment time. An increase in mean annual precipitation corresponding to a vegetation cover of 10%
(i.e. P(V=10%) to P(V=20%); Fig. 5) leads to an increase in erosion rates to a maximum of 0.44 mm/yr after 74.8 kyrs
(factor of 2.2 increase, Fig.9a, blue line). For initial vegetation of V = 70% a vegetation increase of dV = +10% results in
minimum erosion rates of 0.14 mm/yr after 117.7 kyrs (factor of 1.4 decrease, Fig. 9b, green line). A corresponding
increase in precipitation for these same vegetation conditions leads to maximum erosion rates of 0.44mm/yr after 107.5
kyrs which is an increase by a factor of 2.2 (Fig.9b, blue line). The previous results for a positive step change in vegetation
or precipitation demonstrate that the magnitude of change in erosion rates is larger for changes in precipitation rate than
for vegetation cover changes, and in low initial vegetation cover settings (V=10%) the magnitude of change in erosion
rates for changing vegetation is larger (compare green lines Fig. 9a with 9b).


**4.2.2 Negative Step Change in Vegetation Cover or Precipitation**
**Topographic Analysis**
For negative step-changes in vegetation (green curves, Fig. 8d,e,f), the results show a sharp decrease in topographic
metrics associated with shorter adjustment times compared to the positive step change experiments (compare Fig. 8d,e,f
with a,b,c). For step changes in precipitation (blue curves, Fig. 8d,e,f), the increase of topographic metrics happens slower
and therefore with longer adjustment times. A negative step change in vegetation cover from V = 10% by dV = -10%
leads to a decrease of mean basin relief from 269m to 35m, mean basin slope from 11.2° to 2.3° and mean basin channel
steepness from 108 $m^{-0.9}$ to 11 $m^{-0.9}$ which resembles decreases by factors of 7.8, 3.8 and 9.6, respectively. The adjustment
time until a new steady-state is reached is 0.26 Ma (solid green lines, Fig. 8d,e,f). The corresponding negative change in
precipitation leads to an increase in mean basin relief to 512m, mean basin slope to 15.8° and mean basin channel
steepness to 223 $m^{-0.9}$. These increases reflect changes by factors of 1.9, 1.4 and 2.1 with an adjustment time of 4.9Ma
(dotted green lines, Fig.8d,e,f). Mean basin slope results (Fig. 8e) for a step change in vegetation illustrate a pulse-like
feature of initially increasing slope values, followed by a decrease to lower slope values. In contrast, a negative step
change in precipitation induce an initial decrease in slope, followed by a gradual increase in slope to a value higher than
was initially observed before the change.
Simulations with initial vegetation cover V = 70% and dV = -10% show a decrease in mean basin relief from 418m to
356m, mean basin slope from 15.5° to 13.6° and mean basin channel steepness from 172$m^{-0.9}$ to 144$m^{-0.9}$ which resembles
changes by factors of 1.2, 1.1 and 1.2 and an adjustment time of 2.1Ma (dotted green lines, Fig.8d,e,f). Corresponding
negative changes in precipitation lead to increase of basin relief of 465 m, basin slope to 16.4° and channel steepness to
195m$^{-0.9}$ which resembles changes by factors of 1.1 for all three values. Adjustment time in this case is 2.2 Ma (dotted
blue lines, Fig.8d,e,f). Behavior of mean basin slope after the step-change follows the V = 10% simulations but shows
lower amplitudes of basin slope for both step-changes in vegetation and precipitation.

**Erosion Rates**
The positive step-change results (Fig. 9a, b) indicated that erosion rates reach their minimum or maximum with a lag
time, and show significant differences in the magnitude and duration of non-equilibrium conditions depending on if
vegetation or precipitation were changing. Simulations with a decrease from V = 10% to V = 0% (Fig. 9c) show a sudden
increase in erosion rates to a maximum value of 3.5 mm/yr which is an increase from steady state conditions by a factor
of 17.7 which is reached after 19.5 kyrs (green line, Fig.9c). A step decrease in precipitation for this corresponding
vegetation difference (i.e. P(V=10%) to P(V=0%) leads to a  smaller, and protracted (longer adjustment time) decrease
in erosion rates to 0.03 mm/yr after 50.1 kyrs. These conditions cause a factor of 5.6 decrease (blue line, Fig. 9c).
Simulations of V = 70% with a vegetation change of dV = -10% show an increase in erosion rates to 0.27 mm/yr which
is a factor of 1.4 increase after 126.3 kyrs (Fig. 9d). For the corresponding decrease in precipitation the data show a
decrease in erosion rates to 0.15mm/yr after 124.5 kyrs. This resembles change by factor of 1.2 (blue line, Fig. 9d).
**4.3 Transient Topography – Oscillating**

**4.3.1 Oscillating Surface Vegetation Cover, Constant Precipitation**
**Topographic Analysis**
The topographic evolution in simulations with a constant precipitation (10 and 360mm/yr for V=10%, and V=70%,
respectively) and oscillating vegetation cover show a different response than the previous step change experiments. The
differences depend on the initial steady-state vegetation cover prior to the onset of 100kyr oscillations. All observed basin
metrics (Fig. 10) show an initial oscillating decrease in values until a  new dynamic steady-state is reached where the
amplitude in oscillations is less than in the preceding initial adjustment period. Simulations with V = 10% (Fig. 10a) show
a factor of 2.5 decline in the basin relief (from 269 m to 107 m). For simulations with V = 70% the reaction and adjustment
to the new dynamic steady-state is less pronounced with a factor of 1.01 decline in relief (from 410m to 407m) with
positive and negative amplitudes in dynamic steady-state of 1.6 m. While these changes are unmeasurable in reality, they
highlight that for high initial vegetation cover settings that changes in only vegetation cover would be difficult to detect.
Analysis of mean basin slope for the model topographies with low (V=10%) vegetation shows a similar behavior with a
factor of 1.6 decrease of the mean slope (from 11.2° prior to the onset of oscillations, to 6.0°, Fig. 10b). However, before
this new equilibrium is reached, the slopes show an increase in mean slope for the first two periods of vegetation
oscillation which then declines towards the new long-term stable dynamic equilibrium which is reached after
approximately 500 kyrs. Local maxima of mean basin slope coincide with local minima in basin relief. For the V = 70%
simulations, the reaction is significantly smaller with no change in mean slope for the new dynamic equilibrium and
amplitudes of both positive and negative of 0.16°. Mean basin channel steepness (Fig. 10c) reflects the behavior of mean
basin elevation. For V = 10% simulations the mean channel steepness decreases by a factor of 2.7 (from 108 m$^{-0.9}$ to 40
$m^{-0.9}$) with a positive amplitude of 3.7$m^{-0.9}$ and a negative amplitude of 6.1 $m^{-0.9}$. For V = 70% simulations the response is
again only minor, compared to the lower initial vegetation cover simulations with a change of mean channel steepness
from 186 $m^{-0.9}$ to 167 $m^{-0.9}$ and positive amplitudes of 1.1 $m^{-0.9}$ and negative amplitudes of 0.9 $m^{-0.9}$. Like the elevation
data, the steepness data shows a distinct oscillating pattern with a slow increase to local maxima and rapid decreases to
local minima which coincide with maxima/minima of elevation data. Taken together, the previous observations
demonstrate a larger change in topography for oscillations in poorly vegetated areas compared to those with higher
vegetation cover. Furthermore, the magnitude of topographic change that oscillations in vegetation impose on topography
are largest in the first ~500 kyr after the onset of an oscillation, and diminish thereafter.



**Erosion Rates**
The erosion history for simulations with oscillating vegetation cover (Fig. 11) demonstrate large variations in the erosion
rate that depend on the average vegetation cover of the oscillation. Furthermore, pronounced differences in the amplitude
of erosion occur if the vegetation cover is above or below the mean of the oscillation (Fig. 4a). More specifically,
simulations with V = 10% show a pattern of a small decrease in erosion rates (from 0.2 to 0.03 mm/yr) when vegetation
cover increases above the mean cover, in contrast to a large increase in erosion rates (up to 3.3 mm/yr) when vegetation
cover decreases below the mean of the oscillation (Fig. 2, Fig. 11). Maximum erosion rates decline over multiple periods
of oscillation until they reach a dynamic steady-state with maximum rates (1.2mm/yr) at 760kyrs after the onset. Time
periods of higher erosion rates (>0.2 mm/yr) have a mean duration of 28kyrs, whereas periods of lower erosion rates (<0.2
mm/yr) have a mean duration 72kyrs. For simulations with high vegetation cover (V = 70%) the maximum and minimum
erosion rates are 0.28 and 0.15 mm/yr, respectively. The magnitude of maximum and minimum erosion rates are not
significantly time-dependent and are therefore constant over the simulation. The mean duration of periods with higher
erosion rates (> 0.2mm/yr) is 55 kyrs whereas the duration for periods with lower rates (< 0.2 mm/yr) is 45 kyrs. These
results demonstrate that areas with low vegetation cover experience not only larger amplitudes of change in erosion rates,
but also an asymmetric change whereby decreases in erosion rates are lower magnitude than the increases in erosion rates.


**4.3.2 Oscillating precipitation, Constant Vegetation**
**Topographic Analysis**
The evolution of topographic parameters for simulations with oscillating mean annual precipitation and two different
constant surface vegetation covers (V=10 or 70%, Fig. 12) show a less extreme and smaller temporal change in erosion
rate to variations in precipitation compared to the previous discussed effects of oscillating vegetation cover (Fig. 11). In
Fig. 12a, the mean basin relief results for V = 10% and oscillating precipitation show small variations (+4.9 m to -3.8 m)
in relief around a mean of 269 m, which is similar to the mean relief prior to the onset of oscillations at 5,000kyrs. For
simulations with V = 70% the change in relief is slightly more pronounced with factor of 1.1 adjustment to a new mean
(380 m from 418 m) in steady-state conditions. The evolution of topographic slope (Fig. 12b) for V = 10% simulations
shows a factor of 1.05 adjustment to a new dynamic equilibrium (from 11.2° to 10.6°). For V = 70% the mean slope
values do not significantly change from steady-state to transient conditions . Mean channel steepness (Fig. 12c) for

V=10% shows a factor of 1.01 adjustment (from 108 m$^{-0.9}$ to 110 m$^{-0.9}$), and would be difficult to measure in reality. The amplitude of oscillation is 4 m$^{-0.9}$ for both negative and positive amplitudes. For $V_{ini}$ = 70% simulations a factor 1.1 change in channel steepness occurs (from 171 m$^{-0.9}$ to 152 m$^{-0.9}$) with amplitudes of 4.5m$^{-0.9}$ for both positive and negative changes. Figure 12 illustrates changes in topographic metrics that result from oscillations in precipitation occurring around vegetation covers of 10 and 70%. These changes are significantly smaller than those predicted for constant precipitation, but oscillating vegetation conditions (Fig. 10).

**Erosion Rates**

Predicted erosion rates from simulations with constant surface vegetation cover and oscillating mean annual precipitation indicate different amplitudes of change around the mean erosion rate depending on the vegetation cover. For simulations with V = 10% (Fig. 13, blue solid line) erosion rates oscillate symmetrically around the steady-state erosion rate (0.2mm/yr). The maximum and minimum erosion rates of 0.42 and 0.01 mm/yr, respectively, do not lead to a shift in the mean value of erosion rates over time.. In contrast, predicted rates with a higher vegetation cover of V = 70% (Fig. 13, blue dotted line) demonstrate an asymmetric oscillation in rates around the mean, whereby the maximum in rates (0.43 mm/yr) has a larger difference above the mean rate, than do the minimums in the oscillation (0.12 mm/yr). For this higher vegetation cover scenario, a gradual decrease in the mean erosion rate

while time progresses occurs. Furthermore, the maximum and minimum erosion rates decline over several oscillation periods. Taken together, these results indicate that oscillations in precipitation impact erosion with different magnitude depending on the amount of vegetation cover. Areas with low vegetation cover demonstrate the highest and symmetric oscillation of erosion rates due to changes in precipitation whereas in areas with high vegetation cover the effect of negative changes in precipitation is dampened by vegetation.

**5. Discussion**

The previous results highlight different sensitivities to changes in either surface vegetation cover or mean annual precipitation. In the following, we synthesize the previous results and then build upon them to discuss the scenario of synchronous variation in both precipitation and vegetation cover.

**5.1 Interpretation of Steady-State Simulations**

Landscapes in a topographic steady-state show distinct features in topographic metrics that are widely used to estimate catchment-averaged erosion rates and therefore the leading processes of erosion within a landscape (DiBiase et al., 2010). The steady-state simulations presented can reproduce (Fig. 7) variations in topographic metrics over different climate and vegetation states seen in other studies (Langbein & Schumm, 1958, Walling and Webb 1983). Comparison of simulations with homogeneous precipitation and changing values of vegetation cover (Fig. 7a,b,c) to simulations with both changing precipitation and vegetation cover (Fig. 7d,e,f) indicates we can only reproduce a similar trend with a distinct peak in topographic metrics when both variable precipitation and vegetation cover are considered. From this, we conclude that modern model-based landscape evolution studies that aim to compare areas with different climates should incorporate vegetation dynamics in their simulations. Misfits between the predicted and Chilean observed topographic metrics (Fig. 7d,e,f) present when the vegetation and precipitation both vary likely stem from the simplicity of the model setup used and the likelihood of differences of the rock uplift rate and lithology's present in these areas.

**5.2 Interpretation of Step-Change Experiments**

Our analysis shows that changes in vegetation-cover typically have a higher magnitude of impact on topographies for lower values of initial vegetation cover, compared to simulations with high initial vegetation cover (Fig. 8, 9). In those settings the influence of vegetation cover outweighs the influence of precipitation in cases of negative and positive directions of the step change. The reason for this is due to a higher impact of changes in vegetation on erosivity and diffusivity (parameter $K_v$, $K_d$; equation 4, 10) than changes in precipitation.

Furthermore, a negative step change in vegetation cover impacts the topographic metrics by a factor of two more than do positive step change changes (Fig. 8d,e,f). This response is interpreted to be due to the non-linear reaction of diffusivity and fluvial erodibility to changes in vegetation cover (See Fig.6). Negative changes in vegetation cover lead to a higher overall change in diffusivity and erodibility compared to positive step-changes.

Model results for the topographic metrics and erosion rates also indicate a difference in the adjustment times of the system until a new steady state is reached when either precipitation or vegetation cover changes (Figs. 8, 9). For simulations with positive step-changes (Fig. 8a,b,c) the adjustment time for changes in vegetation cover to reach a new equilibrium in topographic metrics or erosion rates is three times longer than the adjustment time for changes in precipitation. Simulations with a negative step-changes in vegetation cover show an adjustment time which is shorter by a factor of 18 compared to negative changes in precipitation. This difference in adjustment time again is a result of the non-linear behavior of erosion parameters $K_d$ and $K_v$ which influence how effective a signal of increasing or decreasing erosion can travel through a river basin (Perron et al., 2012). High values of $K_d$ and $K_v$ are associated with lower adjustment times and are a result of negative changes in vegetation cover. The influence of changing precipitation on adjustment time behaves in a more linear fashion and therefore mostly depends on the overall magnitude of change. Therefore, positive step-changes in vegetation cover decrease $K_d$ and $K_v$ which leads to higher adjustment times than the corresponding changes in precipitation.

An increase and then decrease, or decrease and then increase, in predicted slope and erosion rates is observed for both the positive and negative step changes experiments (Fig. 8b,e; and Fig. 9). This non-linear response in both positive and negative step changes in precipitation and vegetation cover is also manifested in the subsequent oscillation experiments, but most clearly identifiable in the step change experiments. The explanation for this behavior is as follows. A positive step change in vegetation cover (Fig. 8b) leads to a decrease in fluvial capacity because increased vegetation cover increases the Manning's roughness (parameter $n_v$, equation 8). The effect of changing the Manning's roughness varies with the location in the catchment and influences which processes (fluvial or hillslope) most strongly influence slopes and erosion rates. In the upper part of catchments where contributing areas (and discharge) are low, this increase in Manning's roughness causes many areas to be below threshold conditions such that fluvial erosion is less efficient, and hillslope diffusion increases in importance and reduces slopes. In the lower part of catchments, where contributing area and discharge are higher, changes in the Manning's roughness are not large enough to impact fluvial erosion because these areas remain at, or above, threshold conditions for erosion. With time, the lower regions of the catchments that are at or above threshold conditions propagate a wave of erosion up to the higher regions that are below threshold conditions. The propagating wave of erosion eventually leads to increase in slope angles, essential due to the response time of the fluvial network to adjust to the new Manning's roughness conditions.

In contrast, a positive step change in mean annual precipitation leads to an initial increase in fluvial shear stress which initially causes headward incision of river channels and leads to wave of erosion that propagates upstream and increases channel slope values (Fig. 8b, see also e.g. Bonnet and Crave, 2003). The increase in channel slopes leads to an increase

in the hillslope diffusive flux adjacent to the channels that then propagates upslope. Eventually, this increase in hillslope
flux leads to a decrease in hillslope angles, and an overall reduction in mean catchment slopes after the systems reaches
equilibrium.
Negative step-changes in vegetation cover or precipitation (Fig. 8e, green curves) shows the opposite behavior of the
previous positive step change description. A negative step change in vegetation cover leads to an initial increase of fluvial
erosion everywhere in the catchment because the Manning's roughness decreases everywhere. This catchment wide
decrease in Manning's roughness leads to fluvial incision everywhere in the catchment and an increase in mean slope.
However, eventually hillslope processes catch up with increased slopes near the channels and with time an overall
reduction of slope occurs. Negative changes in precipitation (Fig. 8e, blue curves) lead to an initial decrease in fluvial
erosion thereby leading to an increase in the significance of hillslope processes such that slope angles between channel
and ridge decrease as hillslope processes fill in channels. With time, the fluvial network equilibrates to lower precipitation
conditions by increasing slopes to maintain equilibrium between erosion and rock uplift rates.
Thus, the contrasting behavior of either initially increasing or decreasing slopes and erosion rates, followed by a change
in the opposite direction of this initial change highlight a complicated vegetation-climate induced response to changes in
either parameter. This non-linear behavior, and the millenial timescales over which these changes occur, suggest that
modern-systems that experienced past changes in climate and vegetation will likely be in a state of transience and the
concept of a dynamic equilibrium in hillslope angles and erosion rates may be difficult to achieve in these natural systems.
Previous studies have inferred relationships between mean catchment erosion rates derived from cosmogenic
radionuclides and topographic metrics (e.g., DiBiase 2010; DiBiase and Whipple 2011). However, the previous discussion
of how topographic metrics change in response to variable precipitation and vegetation suggest that empirical
relationships between erosion rates and topographic metrics contain a signal of climate and vegetation cover in the
catchment. We illustrate the effect of step changes in climate and vegetation on the new steady-state of topographic
metrics in figure 14. In this example, the new steady state conditions in basin relief and mean slope after a modest (+/-
10%) change in vegetation or precipitation (triangles) differ from the initial steady-state condition (circles). These
changes in topographic metrics when the new steady-state is achieved occur despite the rock uplift rate remaining
constant.
**5.3 Interpretation of Oscillation Experiments**
The results from the 100 kyr oscillating vegetation and precipitation conditions shows that oscillating vegetation cover
without the corresponding oscillations in precipitation leads to adjustments of topographic features, to a new dynamic
equilibrium after approximately 1.5 Ma (Figs. 10, 11). The previously described response of topographic metrics and
erosion rates to oscillating vegetation (see results section) are due to processes described in the previous step change
experiments. For example, the asymmetric oscillations in topographic metrics for V=10% (Fig. 10) are due to the
superposition of positive, then negative changes described in section 5.2. Variations in the imposed Manning's roughness,
and relative strengths of fluvial vs hillslope processes in different parts of the catchments at different times causes the
topographic metrics and erosion rates to have a variable amplitude and shape of response from the symmetric oscillations
imposed on the topography (Fig. 4a).
Simulations with oscillating precipitation and constant vegetation cover however show a less pronounced shift to new
equilibrium conditions and lower amplitudes of oscillation in both topographic metrics and erosion (Figs. 12, 13). This
difference in the response of the topographic metrics and erosion rates shown in figures 12 and 13, compared to the
oscillating vegetation cover experiments (Figs 10, 11), is due to a higher impact of changes in vegetation cover on
parameters which guide erosion rates and therefore adjustment to topographic metrics compared to the \corresponding
changes in precipitation in our model domains (Fig. 5). Especially for simulations with low initial vegetation cover the
effect of changing vegetation has a larger magnitude effects because of the non-linear response of diffusivity and fluvial
erodibility to changes in vegetation cover compared to the linear response to changes in precipitation.
**5.4 Coupled Oscillations in Both Vegetation and Precipitation**
The previous sections present a sensitivity analysis of how step changes or oscillations in either vegetation cover or
precipitation influence topography.  Here we present a step-wise increase towards reality by investigating the topographic
response to changes in both precipitation and climate at the same time. The amplitude of change prescribed for both
precipitation and vegetation is based upon the present empirical relationship observed in the Chilean study areas for initial
vegetation covers of 10 and 70%, and mean annual precipitations for 10 and 360 mm/yr (Fig. 5).  As with the previous
experiments, oscillations in parameters were imposed upon steady-state topography that developed with the previous
values, and a rock uplift rate of 0.2 mm/yr.
Figure 15 shows the evolution of topographic metrics for simulations with combined oscillations in precipitation and
vegetation. The variation in topographic metrics resembles those described for simulations with constant vegetation cover
and oscillating climate by showing little to no significant adjustment towards new dynamic steady-state conditions. The
amplitudes of oscillation are dampened from those of previous results because of the opposing effects of changes in
precipitation and vegetation cover (e.g. compare blue and green curves in Figs. 8 and 9).
However, inspection of the predicted erosion rates (Fig. 16) for the combined oscillations indicates a significant (~0.1; -
~0.15 mm/yr), and highly non-linear response. The response between the 70% and 10% vegetation cover scenarios are
very different such that for heavily vegetated areas (P(V=70%)) erosion rates typically increase during an oscillation,
whereas for the low vegetation cover conditions (P(V=10%)) erosion rates initially show a decrease, and then an increase
and decrease at a higher frequency.
To better understand this contrast in the response to combined precipitation and vegetation changes, the first 100 kyr
cycle is shown in figure 17. After an oscillation starts, the 10% initial vegetation cover simulations show a decline in
erosion rates with the minimum erosion rate correlated with highest values of both vegetation cover and mean annual
precipitation (compare top and bottom panels). This part of the response is interpreted as resulting from the hindering
effect of increased vegetation on erosion rates outweighing the impact of higher precipitation on erosion rates (Fig. 17)
because vegetation decreases the effectiveness of erosion and transport of surface material. After values of vegetation
cover and precipitation start to decline, erosion rates show a very rapid increase to values of ~0.3mm/yr. This increase in
erosion rates is due to an increase in both $K_v$ and $K_d$ (Fig. 3b, equations  4, 5) which outcompetes the effect of precipitation
decrease.
Following this, a sudden drop in erosion rates to 0 mm/yr occurs and lasts for 3 kyrs due to the onset of hyper arid
conditions at minimum precipitation. After this low in erosion rates, they increase again (to 0.3mm/yr) as precipitation
and vegetation cover increase while the effect of increased precipitation outweighs the effect of the non-linear decrease
in $K_v$ and  $K_d$ (Fig. 3b, c; equations 4, 5). Finally, at the end of this complex cycle a decrease in erosion rates occurs (Fig.
17b) while vegetation and precipitation are increasing (upper panel) because the effect of vegetation increases $K_v/K_d$ and
outweighs the effect of increasing precipitation.

Lastly, a clearly different behavior in erosion rates occurs for settings with higher vegetation cover (e.g. P(V=70%), Fig. 17) compared to the previous lower vegetation cover scenarios. As the vegetation cover and precipitation increase (Fig. 17A) in the first half of the 100 kyr cycle, the erosion rates increase (to 0.35 mm/yr). This is due to the increase in precipitation which outcompetes the decline in vegetation influenced erosivity/diffusivity parameters $K_d$ and $K_v$. Following this, when vegetation cover and precipitation decrease in the second half of the cycle, little to no change occurs in the erosion rates. This near static behavior in erosion rates while precipitation and vegetation cover decrease is due to an equilibrium between the negative effect on erosion rates for decreasing precipitation and the positive effect on erosion rates for decreasing vegetation cover.

In summary, the non-linear shape of the vegetation dependent erosivity ($K_v$) and hillslope diffusivity ($K_d$) in combination with linear effects of mean annual precipitation on erosion rates, exert a primary control on the direction and magnitude of change in catchment average erosion rates. Despite a simple oscillating behaviour in precipitation and vegetation cover, a complex and non-linear response in erosion rates occurs. In Fig. 18 we depicted the conceptual end-members of landscape behaviour for the different scenarios of increasing or decreasing vegetation cover and mean annual precipitation for different initial landscapes. The implications of this are large for observational studies of catchment average erosion rates and suggest that the direction and magnitude of response observed in a setting is highly dependent on the mean vegetation and precipitation conditions of the catchment, as well as what time the observations are made within the cycle of the varying vegetation and precipitation. Furthermore, these results highlight the need for future modeling studies (and motivation for our ongoing work), to investigate the response of catchment topography and erosion rates to more realistic climate and vegetation change scenarios, as well as a broader range of initial vegetation covers and precipitation rates than those explored here such that the threshold in behaviour between the two curves shown in figure 17b can be understood.

**5.5 Potential Observational Approaches to Test Model Predictions**

The behaviour discussed in the previous section matches field-data reported by Owen et al. (2010) who analysed soil production rates from bedrock in different climate regimes. This data, under the assumption of steady-state soil thickness, can be translated into denudation rates. They show that for low values of mean annual precipitation, soil production rates vary between 0 m/Ma and 2 m/Ma due to abiotic processes controlling soil production rates. These observations resemble the effect of our simulations with 10% initial vegetation cover, which shows the same variations in erosion rates with intervals of zero erosion rate for hyper-arid conditions (Fig. 17). Areas with higher values of mean annual precipitation show higher values in the soil production rate. These data points were not corrected for different uplift rates in the sample areas so it is not possible to isolate the effect of vegetation/precipitation and tectonic uplift. In general, the observations show no clear isolated trend but more of a cluster of soil production rates among a common mean, situated in a zone controlled by biotic conditions. Compared to our model data for simulations with 70% initial vegetation cover, this resembles the non-intuitive behaviour of an increase in erosion rate for increasing values of vegetation cover and precipitation compared to a constant erosion rate for decreasing values of vegetation cover and precipitation.

Schaller et al. (2018) and Oeser et al. (2018) present millennial timescale (cosmogenic radionuclide derived) hillslope denudation, and soil production, rates from the Chilean (EarthShape) study areas (Fig. 1A) considered in this study. They find the lowest hillslope denudation rates in the arid and poorly vegetated north. Moving south towards higher precipitation and vegetation cover the denudation rates increase until the southernmost location with highest rainfall and vegetation cover where denudation rates decrease again. This non-linear relationship of hillslope denudation rates with

vegetation cover and precipitation is not directly comparable to the results presented here, but is consistent with a) the notion emphasized here that interactions between precipitation and vegetation cover on denudation are non-linear, and b) that the study areas considered here, although tectonically quiescent for tens of millions of years, have varying denudation rates that suggest either variable rock uplift rates, and/or a persistent state of transience in hillslope denudation induced by millennial timescale oscillations in climate and vegetation.

Beyond the previous studies, limited observations are available for comparison to the predictions shown here. The millenial to million-year time scales investigated here can best be evaluated from observations of catchment wide denudation over similar timescales. Cosmogenic radionuclide measurements from modern river sediments offer one means to evaluate these results. Work by Acosta et al. (2015) in east Africa and Olen et al., (2016) in the Himalaya, are also consistent with the results presented here for the range of vegetation cover available in each of these areas. However, the integration time scales that these studies are sensitive to are shorter than what is presented here and prohibit a detailed comparison. A final approach that future studies could pursue is to calculate paleo denudation rate for catchments from a time series of sediments deposits preserved in either lakes (e.g. Marshall et al. 2015) or fluvial river terraces (e.g. Schaller and Ehlers, 2006; Schaller et al., 2016). However, to be most effective, these studies need to target multiple study areas with terrace or lake deposits that span a range of vegetation covers in the upstream catchments.

## 5.6 Model Restrictions and Caveats

Similar to previous work on this topic (Collins et al. 2004, Istanbulluoglu and Bras 2005) , the model setup used in this study was simplified to document how different vegetation and climate related factors impact topography over long (geologically relevant) timescales. We acknowledge that future model-studies should address some of the restrictions imposed by our approach to evaluate their significance for the results presented here. Future work should consider a transport-limited fluvial model or a fully-coupled alluvial sedimentation and transport model. The addition of this could bring new understanding in to how vegetation not only influences detachment limited systems, but also influences sedimentation and entrainment of material. This added level of complexity could however limit (due to computational concerns) the temporal scales over which an investigation like this could be conducted. Future studies could improve upon this work by considering a more in-depth parametrization of how vegetation related processes (e.g. changes in root depth and density, plant functional type) influence topographic metrics and erosion rates. Also, although supported by various publications (Dunne, 1996; Dunne et al., 2010), the assumption of a non-linear response of the effectiveness of diffusional and fluvial processes to increases in vegetation cover has a major impact on the results of these study. A better field-based understanding of these processes and the involved relationships could improve the accuracy of model studies like this.

Also, due to the long-timescales considered here, mean annual precipitation rather than a stochastic distribution of precipitation were implemented. Future work should evaluate how stochastic distributions in precipitation and extreme events in arid, poorly vegetation settings, impact these results, however the long timescale forcings in precipitation and vegetation imposed in this study will likely persit as the background template upon which high-frequency changes are active.

Regarding the vegetation and water-budget, a more sophisticated model of evapotranspiration and infiltration as a function to surface plant cover and plant  functional traits such as rooting depth would improve model predictions and is a priority

for our future work. Improvements will come from planned coupling of surface process model with the dynamic
vegetation model LPJ-GUESS (Werner et al. 2018, this issue).
Our assumption that an increase in surface vegetation cover directly translates to an increase in Mannings roughness is
an additional simplification. The real value of Mannings roughness of a surface will be a function of the fractional
densities of different plant communities per model-patch. We argue that this simplification is however necessary because
it is not possible to know the composition of the plant community for specific areas in our modeled timescales. This could
be resolved by fully coupling a landscape evolution model to a dynamic vegetation model to resolve inter-patch
differences of surface vegetation cover and intra-patch plant functional types.
We also acknowledge that the transient forcings we have chosen for driving our model are simplistic and could be
improved by a higher-fidelity time-series of climate over the last millennia. We choose a 100kyr, eccentricity driven,
periodicity because it is widely recognized that the eccentricity cycles are a main control in driving Earth's glacial cycle
over the last 0.9Ma. While this approach is reasonable for a sensitivity analysis such as we've conducted, it prohibits a
detailed comparison to observations in specific study areas without additional refinement. Our results suggest that a
shorter periodicity, which would resemble other periodicities in the Milankovitch cycle (e.g., 41kyrs, 23kyrs) or shorter
time scale climate variations, such as Heinrich events (see Huntley et al. 2013) would lead to smaller magnitudes of
adjustment to new dynamic equilibria, because of short time spans in high-/low-erosive climate conditions within one
period. Regarding the long time-periods considered, we chose to have a steady-state climate driver in the model without
frequency driven modulation of rainfall events. We argue that over large time scales the occurrence of these events can
be integrated into a meaningful mean value but acknowledge that the incorporation of those events could alter the results
on a short cycle-basis. However because there is no meaningful way to test these frequency distributions against past-
climates, this would add additional unknowns and assumptions into our model parameterization.
Finally, we emphasize that a subset of our results, which resemble small magnitude changes of topographic relief (e.g.,
factor of change 1.01, Section 4.3.1, 4.3.2) are valid results for the predicted synthetic landscapes in our model framework
and not a numerical artefact. However, we acknowledge that these predicted changes are too small to be measured in a
real-world setting.
**6. Conclusions**
The results from our experiments show that the interactions of vegetation cover and mean annual precipitation on the
evolution of landscapes is a complex system with competing effects. Main conclusions which emerge from this study are:
(I) vegetation cover has a hindering effect on hillslope and fluvial erosion but the magnitude on which changes in
vegetation cover affect these processes is a function of the initial state of the system. Changes in systems with higher
initial values of vegetation cover have a less pronounced effect than changes in systems with lower initial vegetation
cover.
(II) In comparison to the Coastal Cordilleras of Chile, the relationship between precipitation and surface vegetation cover
shows a distinct shape: For a 10% increase in surface vegetation cover, the corresponding increase in mean annual
precipitation is smaller in areas of lower vegetation cover and increases for areas with higher vegetation cover. This has
an effect on transient topographies by shifting the equilibrium of vegetation and precipitation effects on erosion rates.
(III) Following our step-change simulations, model results show different behaviours for changes in vegetation-cover and
mean annual precipitation. While increases in mean annual precipitation have an increasing effect on erosion rates and
therefore a long-term negative effect on topographic metrics, an increase in vegetation cover hinders erosion, and leads
to higher topographic metrics. The magnitude of these changes is again dependent on the initial vegetation cover and
precipitation before the step-change.
(IV) Simulations with either oscillating vegetation cover or oscillating precipitation show adjustments to new dynamic
mean values around which the basin metrics oscillate. The magnitude of adjustment is highly sensitive to initial vegetation
cover, where simulations with 10% initial cover show higher magnitudes than simulations with 70% cover, for oscillating
vegetation. Oscillating precipitation leads to lower-/no adjustments but an oscillation of basin metrics around the initial
mean values with generally lower amplitudes compared to simulations with oscillating vegetation cover.
(V) Simulations with coupled oscillations of both vegetation cover and precipitation show only small magnitudes of
adjustments in topography metrics to new dynamic equilibrium similar to simulations with a oscillation in only
precipitation. However corresponding erosion rates show a complex pattern of rapid increases and decreases which results
from a interplay of competing effects of hindering of erosion by vegetation and aiding of erosion by precipitation.
Taken together, the above findings from this study highlight a highly variable behavior in how variations in vegetation
cover impact erosion and topographic properties. The complexity in how vegetation cover and precipitation changes
influence topography demonstrates the need for future work to consider both of these factors in tandem, rather than
singling out either parameter (vegetation cover or precipitation) to understand potential transients in topography.

**Acknowledgements:**
This study was funded as part of the German science foundation priority research program EarthShape: Earth surface
shaping by biota (grant EH329-14-1 to T.A.E. and T.H.). We thank Erkan Istanbulluoglu, Taylor Schildgen, and two
anonymous reviewers for constructive reviews that improved the manuscript. We also thank Associate Editor Rebecca
Hodge for thoughtful handling of the manuscript. We thank the national park service in Chile (CONAF) for providing
access to, and guidance through, the study areas during field trips. Daniel Hobley and the LandLab 'slack' community
are thanked for constructive suggestions during the Landlab program modifications implemented for this study. The
source code used in this study is freely available upon request.

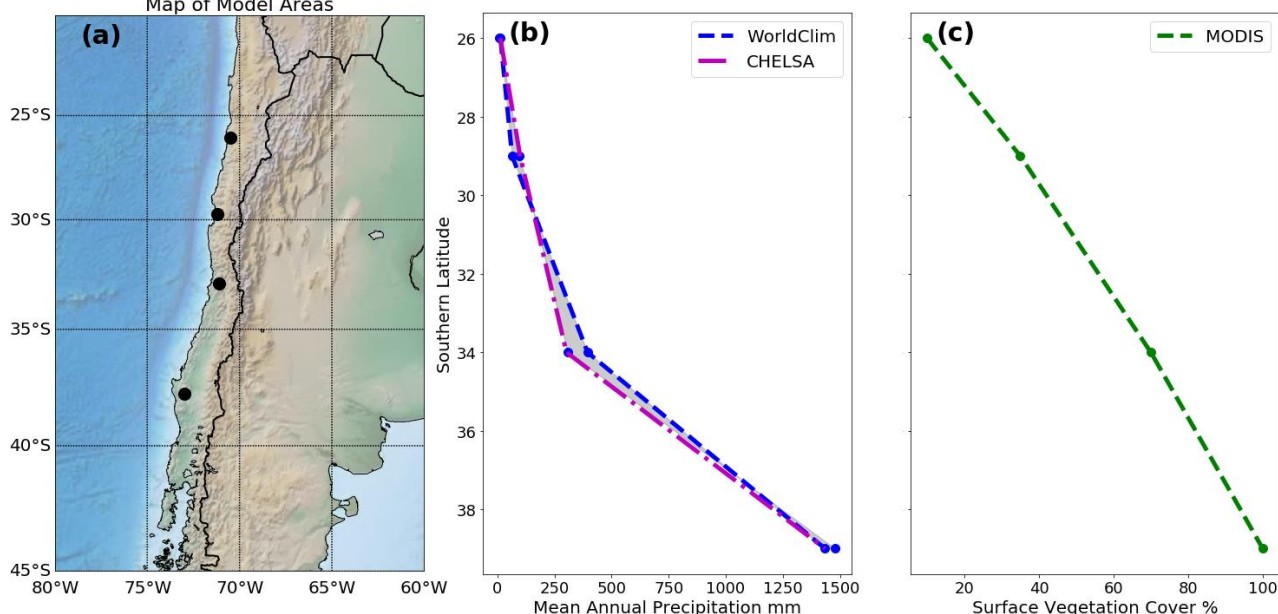


**Figure 1 Overview of the geographic location, precipitation, and vegetation cover of the Coastal Cordillera, Chile studies areas**
**used for model setup up. A) Digital topography of the areas considered and corresponding to the EarthShape**
**(www.earthshape.net) focus areas where ongoing related research is located. B) Observed present day mean annual**
**precipitation from the WorldClim and CHELSA datasets used as model input. B) Present day maximum surface vegetation**
**cover from MODIS data.**

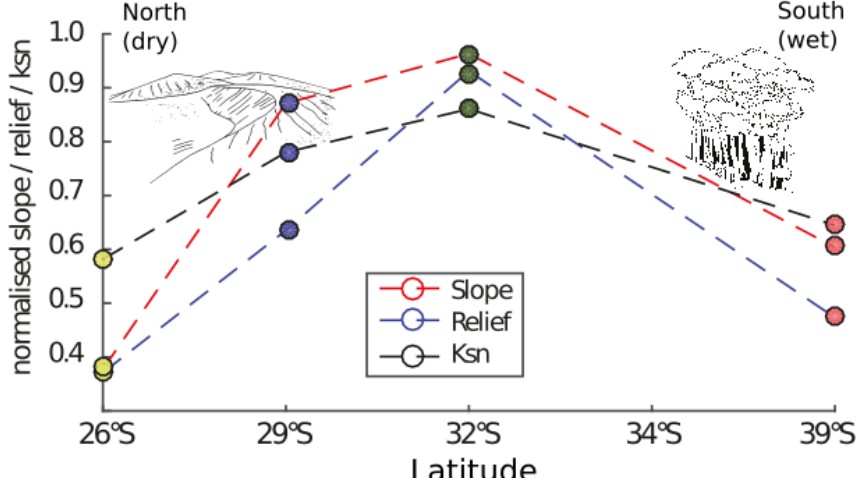


**Figure 2 Normalized basin metrics for study-areas derived from 30 m SRTM digital topography from the study areas shown**
**in Figure 1a. Colored dots represent cumulative mean values of normalized slope, relief and channel steepness calculated for**
**all locations using 5-8 representative catchments in each area. Dotted lines represent linear interpolation between values. Note**
**the gradual increase, then decrease in all metrics around study area at ~32°S.**

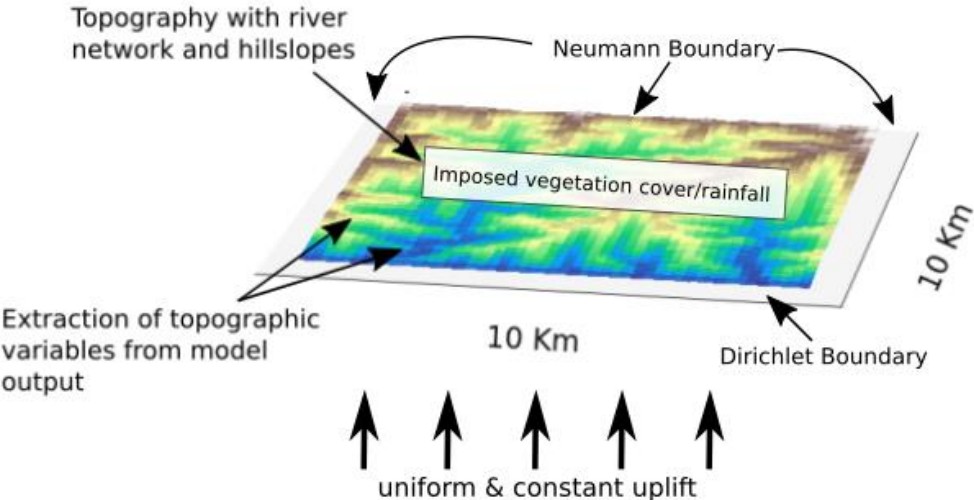


**Figure 3 Example model setup used in simulations in this study. Figure shows an example model predicted topography with a**
**set drainage network, draining to the south. Boundary conditions and parameterizations used in the models are labeled. Blue**
**colors represent low elevations, brown colors represent higher elevations. Additional details of parameters used are specified**
**in Table 1.**

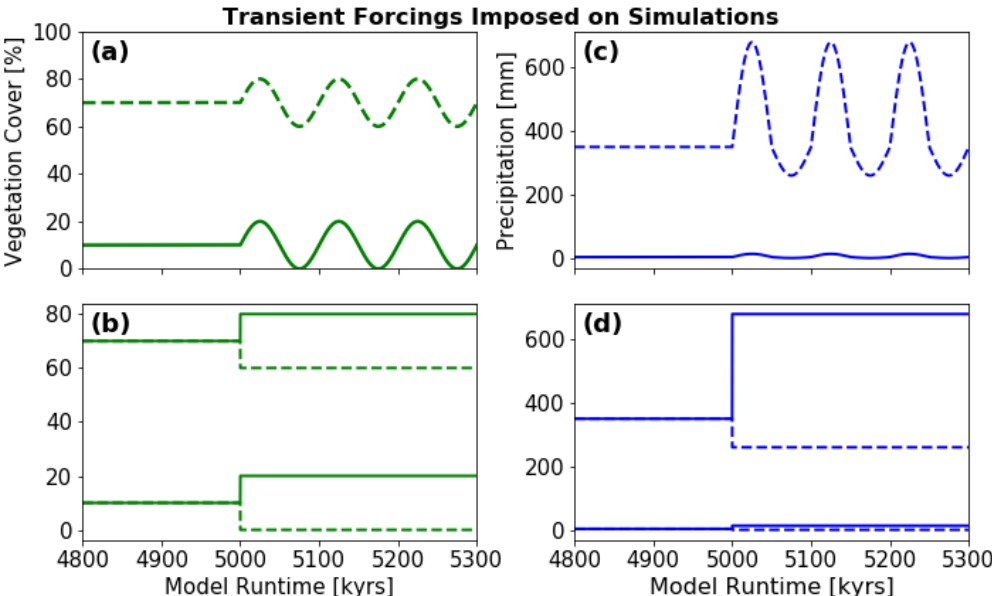


**Figure 4 Transient forcings in vegetation and precipitation considered in model experiments. Simulations were run for 15 Myr**
**prior to the runtime shown in the figure. All transients imposed started a runtime of 5 Myr. A) Variations in vegetation cover**
**imposed in the oscillating experiment conditions for initial vegetation cover of 10 and 70%. Oscillations have a 10% amplitude**
**and a 100kyr periodicity. B) 10% positive and negative step-changes in vegetation cover imposed on simulations with 10% and**
**70% initial vegetation cover.C) Oscillating mean annual precipitation. Positive and negative amplitudes of oscillation resemble**
**the magnitude of precipitation change extracted from vegetation cover/rainfall relationship from satellite data (Fig. 5, D)**
**Positive and negative step changes in mean annual precipitation. The initial precipitation was based on values extracted from**
**the worldclim climate dataset for respective focus areas.**

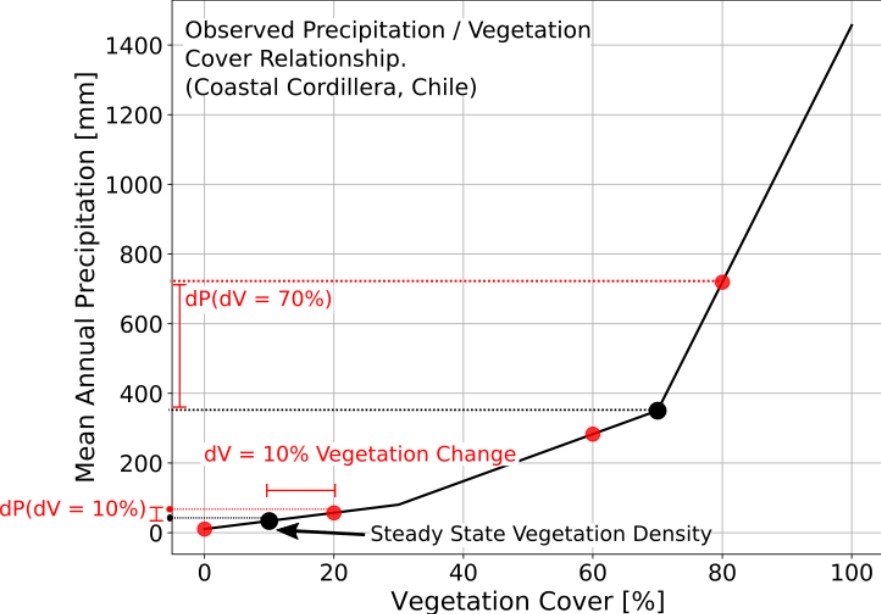


Figure 5 Graphical representation of the observed precipitation – vegetation relationship in the Chilean focus areas (Fig. 1) and how precipitation amounts were selected when perturbations in vegetation cover were imposed. Black dots represent vegetation-precipitation values used in the steady-state model conditions and prior to any transients. Red dots show how vegetation cover perturbations in +/- 10% in the model simulations were used to select corresponding mean annual precipitation amounts. Note that the observed relationship between observed precipitation and vegetation cover in the Coastal Cordillera of Chile is non-linear, and is a source of the non-linear behavior in model forcing (e.g. Fig. 4) and results (Fig. 17) presented here.

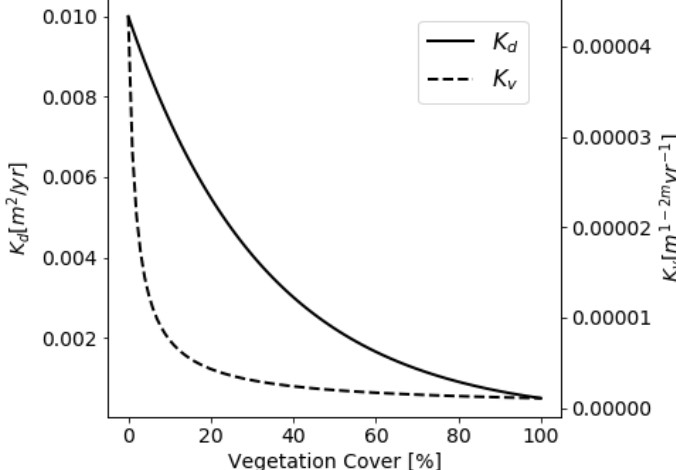

784

Figure 6 Predicted values of hillslope diffusivity $K_d$ (solid line) and fluvial erodibility $K_v$ (dashed line) as a function of vegetation surface cover. Although absolute values can't be compared due to different units, the shape of the curves representing the different parameters show different sensitivities to changes in vegetation cover, and major source of the non-linearities discussed in the text. Fluvial erodibility shows the highest magnitude of change for vegetation cover values < 25% whereas hillslope diffusivity reacts in a more linearly with highest change below < 65% vegetation cover.





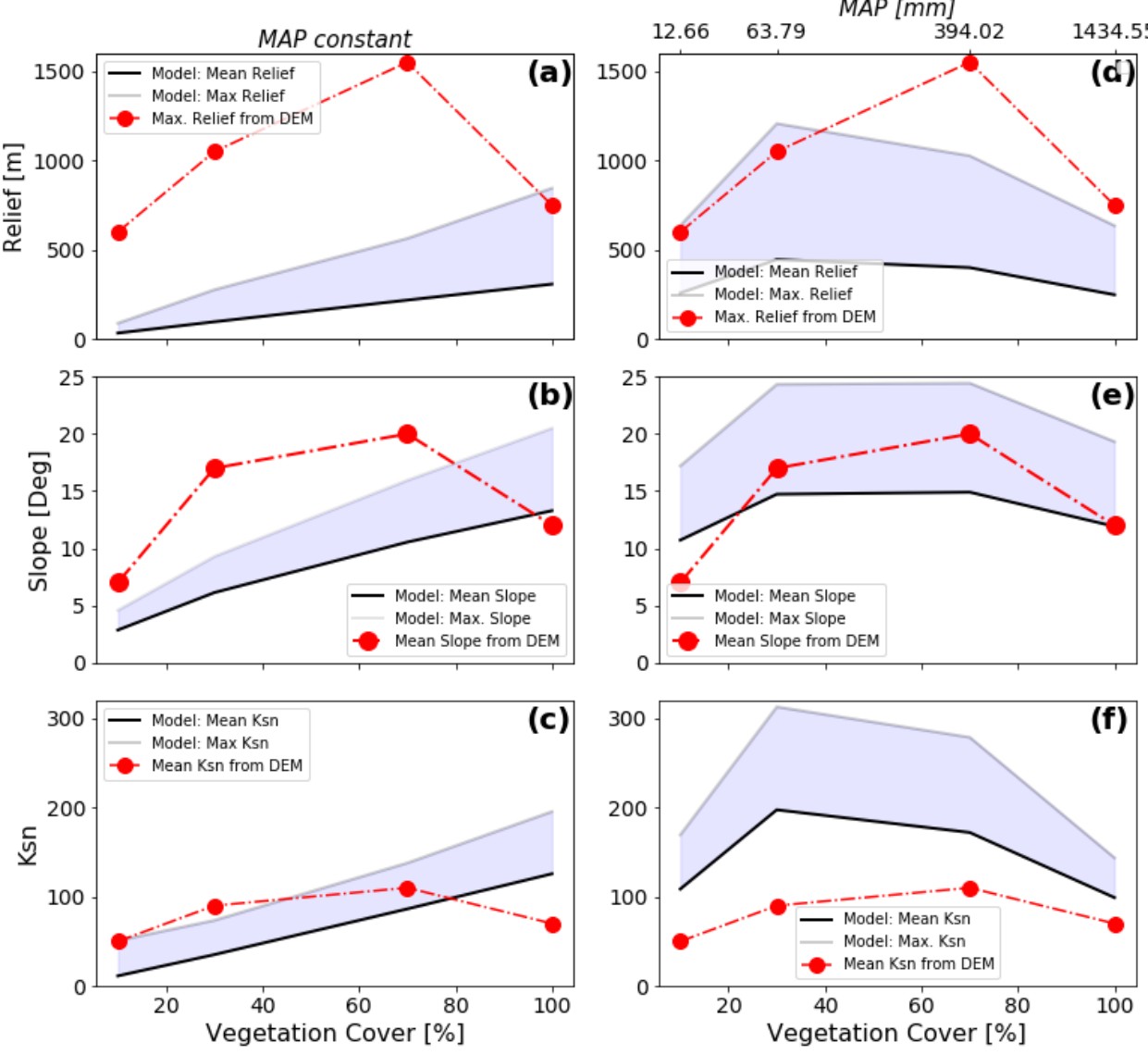


**Figure 7 Steady-state model predicted (shaded regions) and observed (red dots) topographic metrics from the study areas**
**shown in Figure 1 for different vegetation cover amounts. Observed topographic metrics were extracted from SRTM 90 m**
**DEM. Model predicted values are shown for the cases of constant mean annual precipitation (a,b,c) or variable precipitation**
**(D,E,F). Variable precipitation rates and vegetation covers were selected for these simulations using the observed values from**
**the focus areas (Fig. 5). Note that for variable precipitation and vegetation cover simulations (d,e,f) the predicted values (similar**
**to the observations) develop a humped shape pattern of an increase and then decrease in each parameter suggesting the changes**
**in both precipitation and vegetation cover are needed to reproduce the general trend seen in observations. The sources of misfit**
**between the predicted and observed values are due to the simplified (and untuned) setup of the simulations and discussed in**
**the text.**

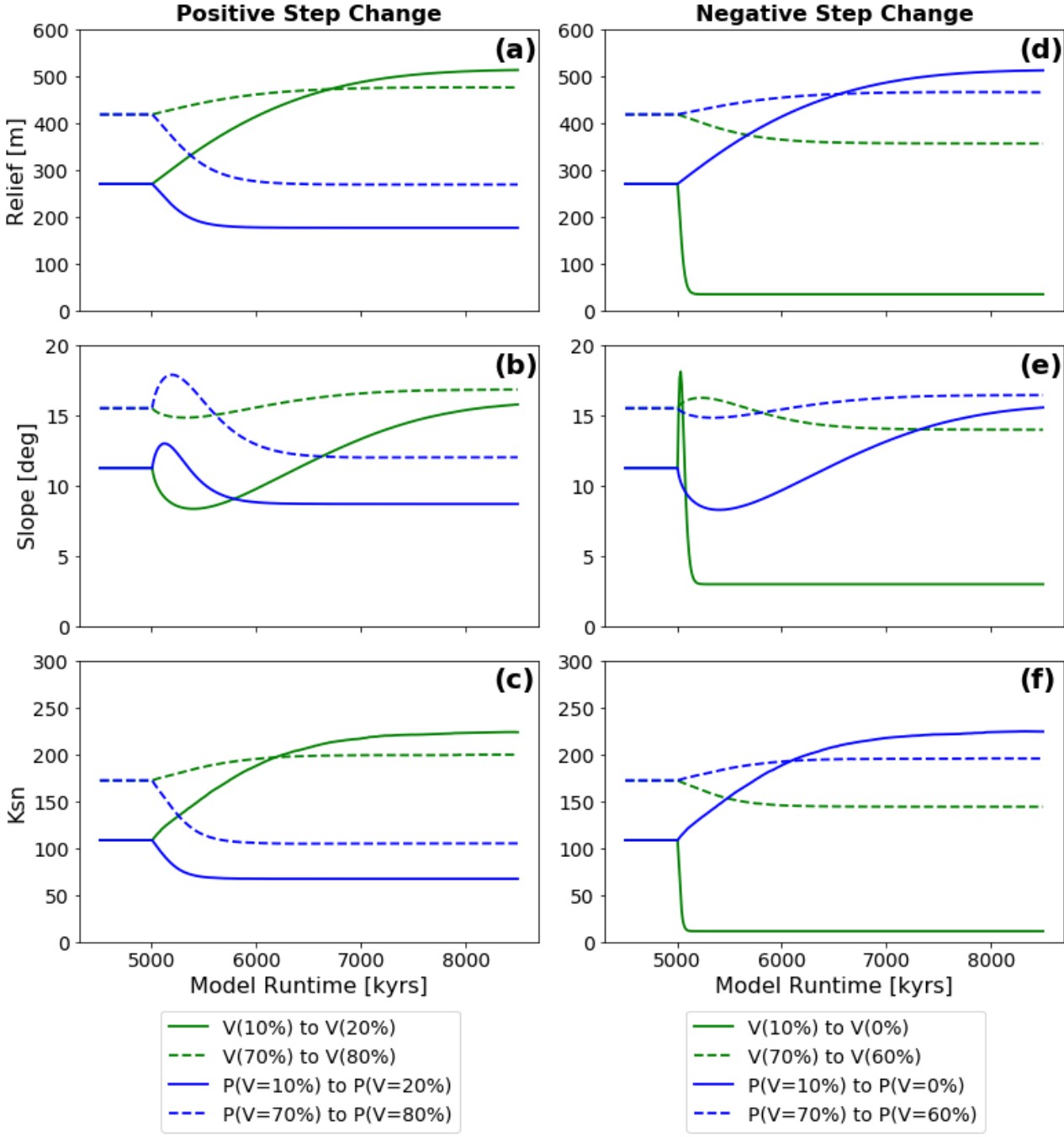


**Figure 8 Observed evolution of topographic metrics after a step-change in either vegetation (green lines) or mean annual precipitation (blue lines). Results are shown for two different initial vegetation cover amounts of V=10 and 70%. Imposed mean annual precipitation changes were done by selecting the precipitation amount corresponding to the initial and final vegetation amounts used in the simulations for vegetation cover 'only' change. Panels a,b,c show the reaction of model topographies to positive changes in boundary conditions, panels d,e,f show the reaction to negative changes in boundary conditions.**



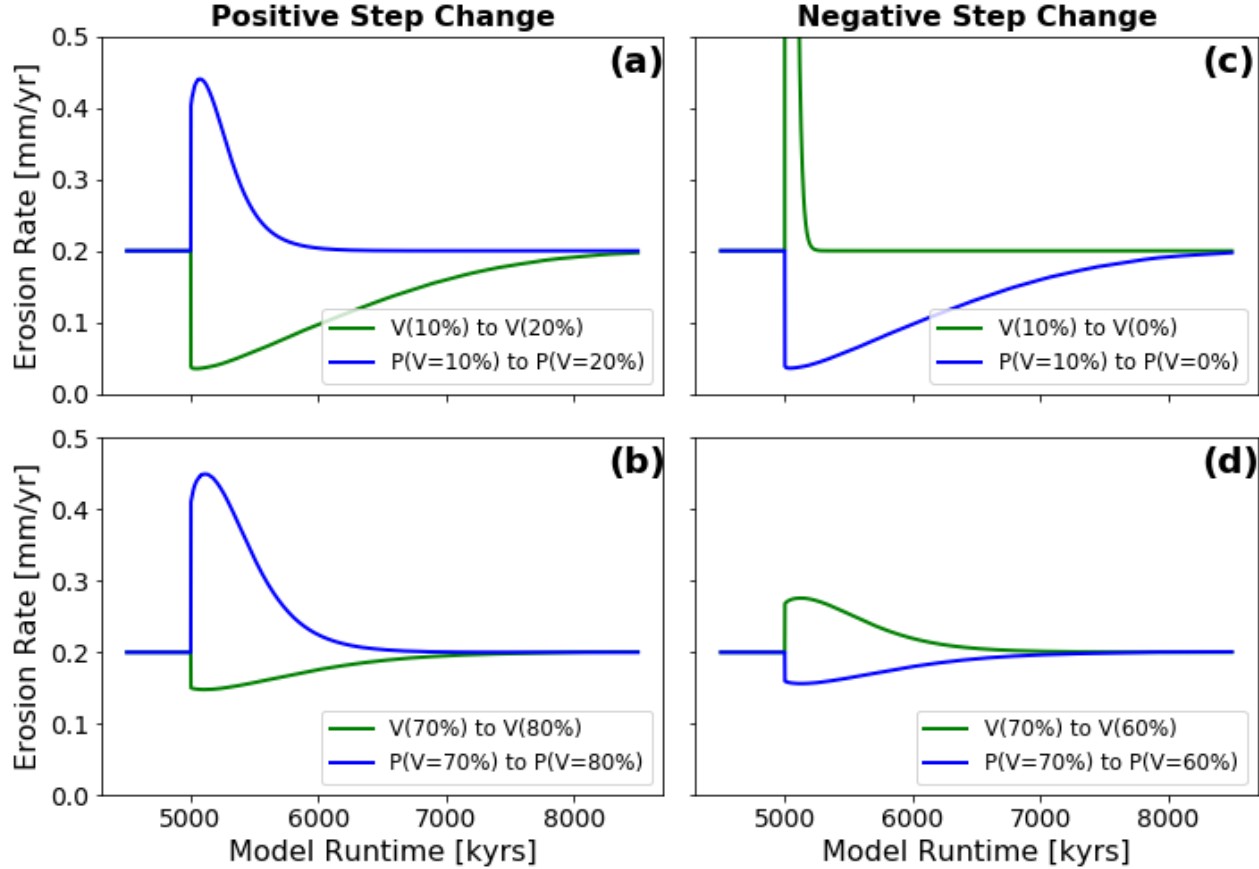


**Figure 9 Mean catchment-wide erosion rates after a step-change disturbance in model boundary conditions. Blue lines**
**represent erosion rates for models with changes in only precipitation, green lines represent erosion rates for models with**
**changes in only vegetation cover. Panels a,b show the evolution after positive step-change, panels c,d for models with negative**
**step-change. Note that the direction of change (positive or negative) from the initial state is in the opposite directions for**
**precipitation and vegetation cover changes. This effect is manifested in the subsequent plots.**

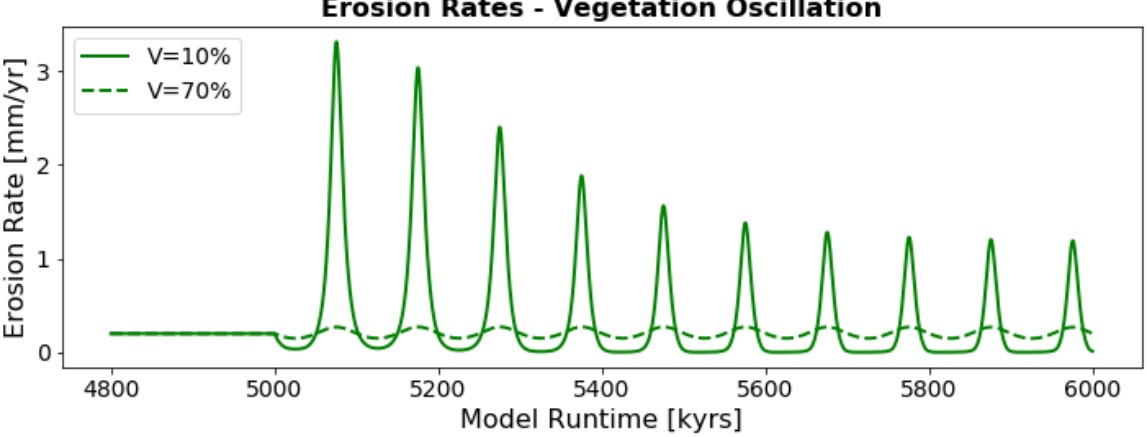


**Figure 10 Evolution of topographic metrics for simulations with oscillating surface vegetation cover and constant precipitation**
**corresponding to the initial vegetation cover prior to the transient in vegetation cover. Panels a,b,c show mean basin relief,**
**mean basin slope and mean basin channel steepness ($k_{sn}$), respectively.**

**Figure 11 Predicted mean catchment erosion rates for simulations with oscillating surface vegetation cover and constant**
**precipitation. Note that the magnitude of change in erosion rates for +/- 10% change in vegetation covers differs depending on**
**the initial (or background) vegetation cover. This non-linear response is due in part to the vegetation cover effects on rock**
**erodibility and diffusivity shown in figure 6.**

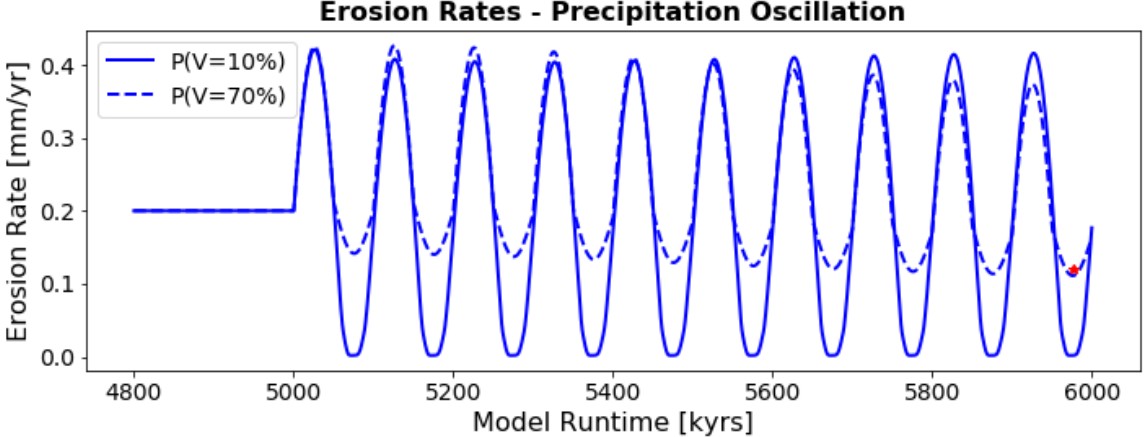


**Figure 12 Evolution of topographic metrics for simulations with oscillating mean annual precipitation and constant vegetation cover. The vegetation cover was held constant at the value corresponding to the precipitation rate prior to the onset of the transient at 5000 kyrs. Panels a,b,c show mean basin relief, mean basin slope and mean basin channel steepness ($k_{sn}$), respectively.**

834

**Figure 13 Mean catchment erosion rates for simulations with oscillating mean annual precipitation and constant surface vegetation cover. The amplitude of change in the erosion rates varies with the initial vegetation cover, in part due to the non-linear relationship between precipitation and vegetation cover (Fig. 4, 5).**

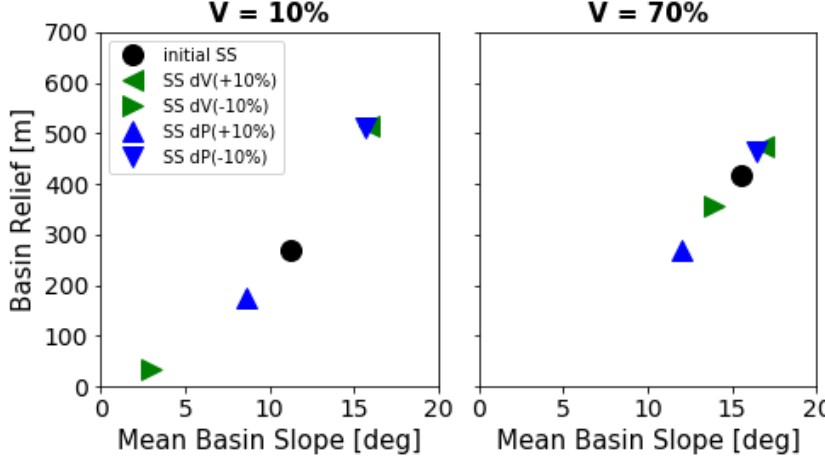

838

**Figure 14 Shifts in mean basin slope/mean basin relief relationship for simulations with positive and negative step-changes in either vegetation cover (green triangles) or mean annual precipitation (blue triangles). Black dots represent initial steady-state conditions prior to any imposed transient in vegetation cover or mean annual precipitation. Note that the sensitivity of topographic relief to perturbations in precipitation or vegetation cover is highest for low-vegetation cover (10%) settings.**

843

## Combined Vegetation and Precipitation Oscillation

844

**Figure 15 Evolution of topographic metrics for coupled simulations where both changes in surface vegetation cover and a corresponding change (Fig. 5) in mean annual precipitation are simultaneously imposed. The amplitudes and frequency of the forcings that were imposed on the simulations are the same than the ones used for the simulations with isolated transient forcings. Panels a,b,c show evolution of mean basin relief, mean basin slope and mean basin channel steepness ($k_{sn}$) after start**

of oscillation at 5Ma. Note the muted/damped response relative to previous simulations of oscillating vegetation cover or
precipitation conditions.

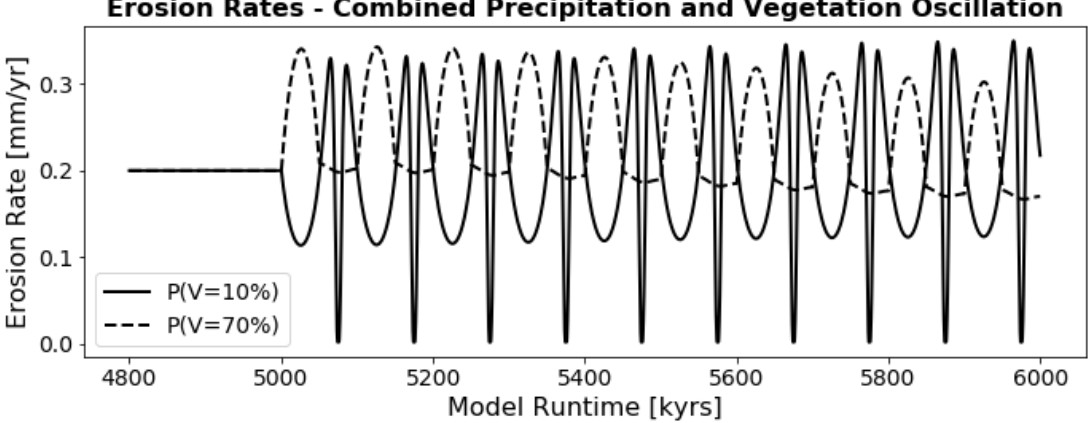


**Figure 16 Mean catchment erosion rates for coupled simulations with changes in surface vegetation cover and mean annual**
**precipitation. The first cycle in the time series is expanded in Figure 17. The variable amplitude and non-linear response shown**
**here is due to the combined non-linear forcings in precipitation (Fig. 4, 5) and rock erodibility and diffusivity (Fig. 6) for**
**different initial vegetation cover amounts.**

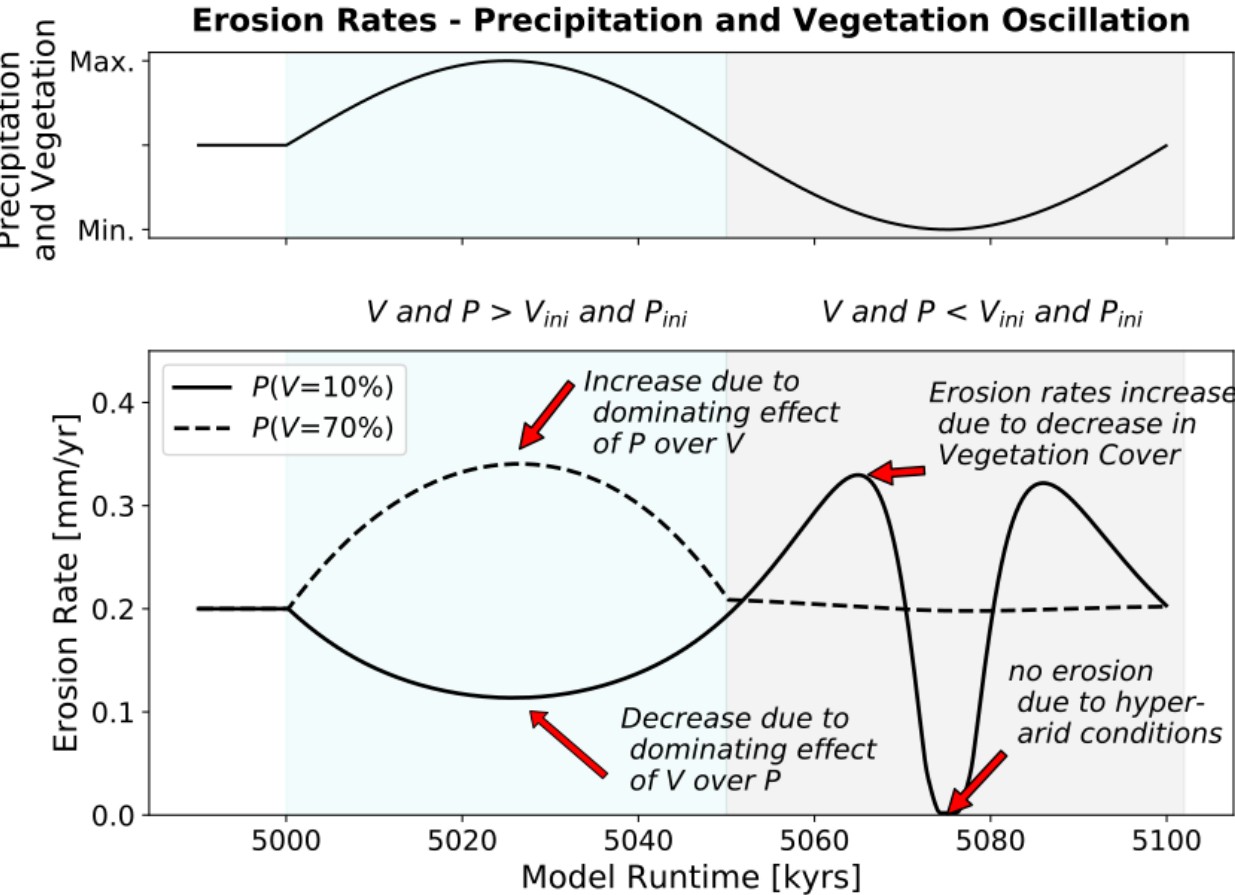


**Figure 17 Mean catchment erosion rates for coupled simulations for one period of oscillation after the start of transient**
**conditions (see also Fig. 15). Upper subplot shows conceptualized transient forcing in vegetation cover and mean annual**
**precipitation, lower subplot shows erosion rate for simulations with low (black line) and high (dotted line) initial vegetation**
**cover and precipitation values.**

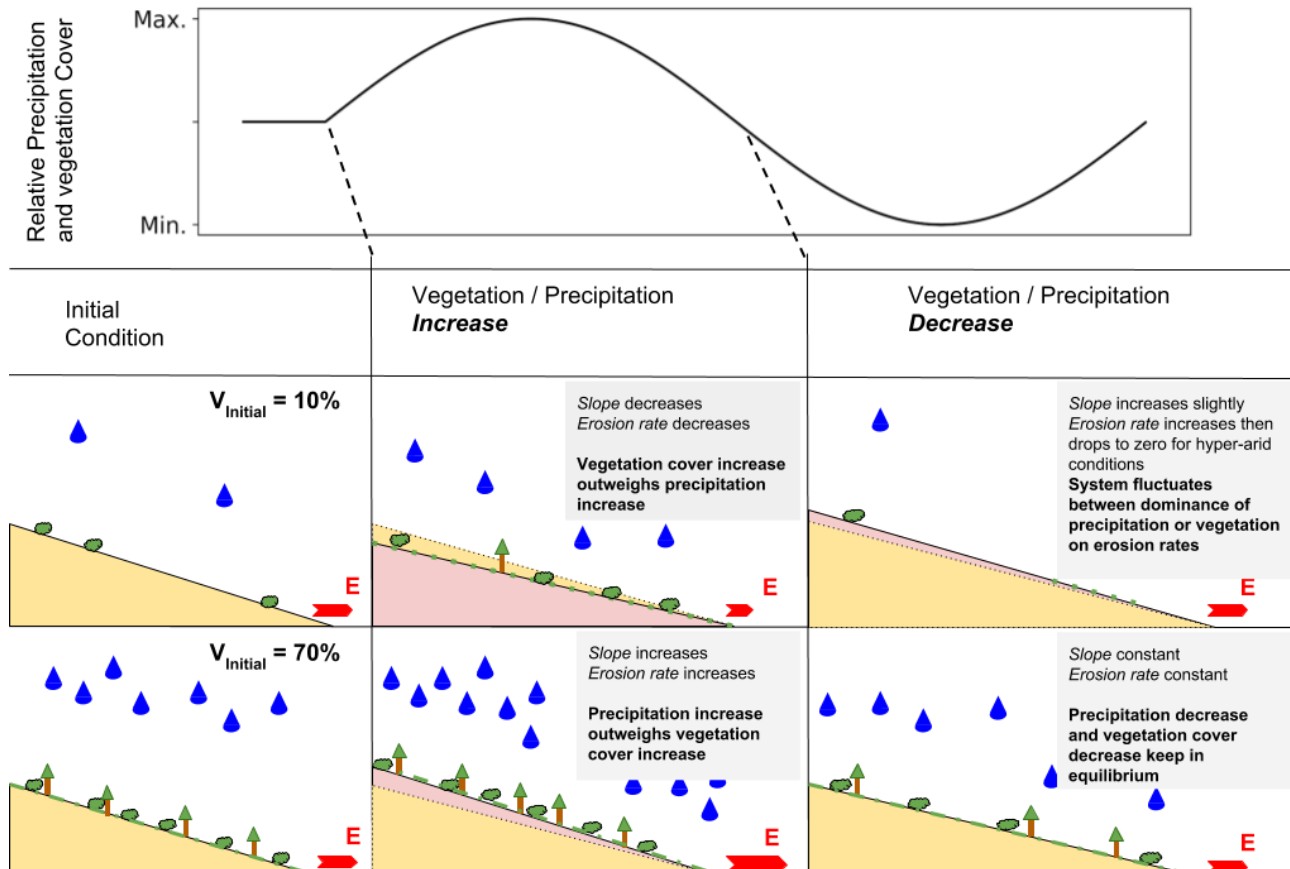


**Figure 18 Conceptual figure showing the topographic response from simulations with coupled oscillation of mean annual
precipitation and vegetation cover (see Fig. 17 for erosion rates). Upper row illustratestransient forcings, lower two rows shows
initial topography (yellow) and resulting transient topography (pink). Changes in topography are not to scale. Vegetation and
rainfall amount is shown qualitatively on the hillslopes.**


**Table 1** Model parameters used for Landlab model setup.

| Model Parameter | Unit | Value |
|---|---|---|
| Uplift (U) | mm/yr | 0.2 |
| Fluvial Erodibility (ke) | m/yr (Kg m1 s-2)-1 | 7.00E-06 |
| Critical Shear Stress (\tau c) | Pa | 58 |
| m, n | - | 0.6 / 0.7 |
| Base Diffusivity (Kb) | m2/yr | 0.02 |
| Mannings Number (Vegetated, nvr) | - | 0.6 |
| Mannings Number (Soil, ns) | - | 0.01 |
| Reference Vegetation Cover (Vr) | - | 100% |
| w | - | 1 |
| alpha | - | 0.3 |
| p | - | 1 |
| Transient Vegetation Cover Amplitude (+-dV) | % | 10 |

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
