# Peer review of "Effect of changing vegetation and precipitation on denudation"

_Earth Surface Dynamics, 2018_

## Referee Comment (RC1) · Anonymous Referee #1 · 5 Mar 2018

Summary: The authors attempt to link changes in landscape form to changes in climate and vegetation. Although this is a worthy topic to explore, it would be a challenge for even a more complicated model because we simply don't know enough about the processes and feedbacks involved. Considering that this model greatly oversimplifies what we actually do know, its contribution to our understanding is not obvious. Those familiar with these processes will view the results with skepticism, and those who aren't may believe the results without fully appreciating all of the short-cuts and assumptions baked into the governing equations. I know how much work goes into

modeling exercises like this so I always try to be open-minded when reviewing these types of manuscripts but, in this case, I cannot recommend publication.

Main comments

My comments are mainly focused on the governing equations. Because I believe them to be either unsupported or flawed, I don't address the results in detail. If the governing equations of a model are not honoring reality in some fundamental way, then its output will be unreliable.

It didn't seem like the authors tried to determine whether the model was working correctly. Thanks to 10Be, we have erosion rates for many watersheds around the planet, including catchments that are similar to the ones modeled here. Before we can believe that the model works, the authors ought to run it on one with published erosion rates. This would have been an important first step before embarking on the rest of the project.

This model is driven by, essentially, two governing equations. The first describes soil creep via linear diffusion. The authors use a formulation, proposed in a paper from 2005, that links the diffusivity to vegetation density that was based on little data and only accounts for physical processes (eg, rainsplash) and ignores bioturbation. Are there field observations to support that physical processes dominate soil creep at their field site? Importantly, there is no support for the nonlinear equation that relates diffusivity to vegetation density. Over a narrow range of precip, Ben-Asher et al (2017) found a linear inverse relationship between diffusivity and precipitation which, when combined with the present study's relationship between veg and precip might yield something like the negative exponential equation adopted here but the authors have not demonstrated that. Moreover, a paper that examined diffusivity across a wider range of precip (Hurst et al, 2013) found the relationship to be weakly positive – which runs counter to what the authors have assumed. There is little support, then, for the way the soil creep equation has been parameterised.

The second governing equation describes erosion by flowing water. Before describing my main concerns, I should point out that this section (lines 193-215) was difficult to follow and was missing some critical details. For example, variables appear without explanation or description and the relationship between eqn 5 and the others that follow was unclear. Also, there was no explanation of how rainfall is applied (eg, storm frequency and magnitude), how runoff is generated, or how runoff generation is affected by vegetation density (eg, via interception). The point about storm frequency and magnitude is especially important because changes in climate will affect the distribution of both of these but not necessarily in a uniform way. Given that, here are my main comments regarding the way that erosion by flowing water is treated in the model.

1) Linking the roughness coefficient to the vegetation in the way that was done here ignores the fact that the effect of vegetation goes beyond a simple measure of 'vegetation density.' For example, imagine two landscapes with 70% vegetation density: one is covered by shrubs such that the ground surface between each plant is essentially bare while the other is covered by grasslands. These two landscapes, despite having the same vegetation density, will have different manning's n values on the hillslopes. Since we know that vegetation community changes with climate, the model's attempt to scale manning's n on the basis of vegetation density is not realistic. Indeed, I looked at the field sites via Google Earth and it was clear that vegetation community does change as a function of precip in those regions. I can easily imagine situations where manning's n actually increases with a decrease in vegetation density, the opposite of what is assumed here.

2) Given the comments above, both landscapes will also have different critical shear stresses. For example, soil with shallow grass roots will be more difficult to erode than the bare soil between shrubs. It doesn't appear that the model takes this into account.

3) It appears that the model doesn't distinguish between overland flow on hillslopes and river flow. If so, the authors are assuming that a source of roughness on the hillslopes – the vegetation – is also contributing to roughness in the rivers. For example, if the

authors envision shrubs growing on their hillslopes then they must also be growing in the rivers. Again, I don't see how this is realistic. Moreover, the type of vegetation really matters here with respect to flow depth. Short grasses would have a greater manning's n with low flow depths (ie, overland flow) than with deeper flows (ie, rivers). Conversely, shrubs would have a lower manning's n with overland flow than with river flow.

4) There was no explanation of how the critical shear stress was calculated. Presumably some assumptions were made regarding bed and hillslope material but these were not described. Does the model keep track of the evolving particle sizes as climate changes? More vigorous runoff will coarsen the river beds and hillslope surfaces but I didn't get the sense that this was incorporated into the model. Also, a lower vegetation density will expose the ground surface to raindrop impacts that will mobilize finer material more readily. Was this accounted for? Again, it didn't seem like it.

One of the model's limitations are well-illustrated by Figure 9. It predicts long-term erosion rates on the order of about 0.2 mm/y. This is on the high end of known soil production rates; I would venture to guess that soil production rates at the sites in Chile are quite a bit lower given the dry conditions (0.2 mm/y is what you get in weak-ish bedrock in the Oregon Coast Range where its wet and has lots of trees doing physical weathering). This means that, at these high erosion rates, the landscape would run out of soil yet the model seems to assume an inexhaustible supply of erodible material. In the real world, the loss of soil would have important consequences for runoff processes and the ability of plants to grow but the model seems blind to these.

Finally, there was no attempt to provide any error estimates in the predictions. I understand that this is not common practice with landscape evolution models but it should be and can be done (see papers by Tom Dunne on stochastic modeling). For example, the model makes certain predictions about how erosion rates may vary over time after changes in vegetation (Figure 9). Given all the potential uncertainties embedded in the governing equations and how they were parameterized, how confident are the authors that a predicted erosion rate of 0.2 mm/y is statistically different from a predicted erosion rate of 0.4 mm/y? I'm skeptical that this model can predict annual erosion rates accurately to one tenth of a millimeter.

---

## Short Comment (SC1) · 18 Apr 2018

This is a very interesting paper. I have some suggestions and general comments for improvement and clarity of the theory used..

The length of the paper can be reduced. I felt some descriptions were repetitive under various subheadings which can also be reduced.. It follows a fairly standard writing style that described methods, results, discussion etc..but in each of these certain methods are repetitive. For example 4.1, 4.2..are part of the results section but there are paragraphs that still tell what was plotted w/o giving results. In the discussion of the model results quantitative details were presented. Given the model was not calibrated

for the study watersheds, I wonder if those details matter or realistic at all. This study could lead to more qualitative results that can be summarized/discussed in a conceptual model. But in order to do that the number of simulations done may not be sufficient. My main concern is that the results presented here for constant and transient changes individually P and V and combined, is only one potential out come of a wider range of responses. The paper generally does not ask the "why" question in presenting the various results, but rather literally reports the model results in terms of modeled erosion rates etc.. I wonder if the authors can think of presenting a conceptual model to explain and summarize the various model results.

What was the basis of using 10% and 70% V in the model simulations. I might have missed it. It sound like for a given mean annual P, you need a mean annual V for the sensitivity analysis of the model.

The parameter selection was not sufficiently developed. The Manning's roughenss for bare soil is very low for overland flow. What was nV. Kv is very sensitive to n, and keeping n as low as reported in the paper will increase the sensitivity of Kv to low values of V..

The interplay between vegetation and precipitation on erosion rates and landscape evolution is the relatively novel aspect of this paper. I don't think this was discussed in earlier papers, especially by separating V and P scenarios and then combining them based on the dependence between V and P. The one comment I have on the reconstruction of P and V is that, the paper relates P to V as far as I can tell (Fig. 5). I have a hard time rationalizing this as clearly V responds to P, rather than the opposite. I wonder if the results, especially for the last case where a complex response as observed, would be any different if V was predicted from P, and P oscillated using a sin function.

The paper is long. The authors report details about model results as V and P varied. Details such as the rate of erosion etc can be omitted as these are apparent in the plots and in a theoretical study that does not claim to represent a certain region the exact

rates of erosion would probably not interest the reader. I suggest focusing more on the conceptual findings of the paper. To that end, however the model cases considered seems to be limited. The paper describes very interesting responses of erosion and landscape evolution driven by changes in vegetation and precipitation, separately and in combination. However given this theory it would be important to discuss when these cases occur.. For example, given the complex response presented in the last scenario, I wonder how plausible is the modeled complex response, are there any observational evidence on this in super arid regions?

Important limitation to realize is that vegetation is spatially uniform, and it does not have seasonality in response to radiation and weather. Rainfall is also seems to be steady state although it was not mentioned in the paper. Such variability, if included, can effectively lead to crossing of erosional thresholds with certain frequency. The reason I'm bringing this up is that the model shows a muted response to Vegetation oscillations for V=70%. While this is high veg cover and the variability may be less in the erosional response, however absence of seasonal veg dynamics and stochastic rainfall and vegetation loss due to scour might play a strong role in stabilizing the land. If these above mentioned processes were used, with steeper equilibrium slopes under V=70%, the model would have responded in some episodic fashion.

Line 69: Yetemen et al., 2015a, b are also exceptions to this statement as these papers represented daily water balance, runon-runoff, distributed energy balance and evepotranspiration and transient vegetation growth.

Lines 160-162, a citation would be great here on the use of 100 kyr cycles.

Lines 163 – 169: Were the results vegetation dynamics model results with cyclic climate not used.? Perhaps the actual data used to construct Fig 5 may be shown. I also noticed that in Fig 5 while Veg cover oscillates following what looks like a sinusoidal curve, the Precip data oscillate differently. Does lines 167-168 explain the reasons for this, which I was not sure if I understood correctly. For each 10% in veg cover you

change the % in precip. But was this assumption concluded from Fig1,which would give us % V change for %P change within the ranges of P used in the model ? Given precip drives vegetation shouldn't logically Precip be following the sin curve rather than Veg, unless there is a better explanation and rational telling the reader why the curves were plotted differently.

I could not find where 10% constant vegetation and near zero precip was mentioned.

Some more details on the vegetation cover and the erosional history of the region would be good to include. For example are there any studies that quantified the erosion rates in the Holocene in this region.

Equation 1: please change slope to curvature in the hillslope diffusion term. In the text below the equation kdS is correctly defined as flux but the equation does not use flux it uses the divergence of this flux that leads to change in elevation [L/T] and therefore curvature instead of slope is used. Also if you use curvature in this form the sign in front of the hillslope diffusion component of elevation change should be positive as slope would need another negative sign when represented by elevation change.. BTW equation 2 is correct.

Equation 5, Is the Ethreshold used in the model experiments?

Equation 8 was given incomplete. It's missing the shear stress partitioning part that gives the drop of effective shear stress with V, which is correctly plotted in Fig 6. See equations 10 and 11 in the cited paper in this section.

In this model how is rainfall incorporated.. The rate and duration of rainfall would influence the selection of the threshold to make the model results realistic.

Line 220: please tell what this steepness index was and provide citation..Did you extract the channel network to calculate this?

Line 221: Fig 2captions says 90 m DEM used.

Fig 7 d, e, f.. Lines 259.. In calculating channel steepness how were channels extracted? Here the authors make comparison between model predictions and the actual landscape.. Earlier the reader was informed that the model was not specifically calibrated for this region and the purpose of the study is to investigate the model sensitivity to V and P and compare the general trends between observations and models. Here the model is compared directly with actual data values from DEMs and the authors point out under and over estimations of the model..

Given that the model was not calibrated these statements undermine the strength of the paper. This model have enough parameters to calibrate with which the Fig 7 d,e,f can look a lot better.. This brings a few questions on model parameter selection. For example how was erodibility selected? I presume m and n exponents are also constant. Manning's coefficient for full vegetation is very low and unrealistic. This value is more smooth concrete channel value. Can the authors elaborate if any calibration at all was attempted? If the authors want to stress on the general patterns predicted by the model rather than a poor direct comparison between model and observations, they can report these figures in a non-dimensional form so that the amplitude of responses are compared with respect to 1 in both model and data. A simpe way to non-dimensionalize would be to dive all the values with the mean (slope, relief etc.).

Fig 8 and 9 results make sense..

Fig 13. This figure shows that for the case of denser vegetation cover (V=0.7 or 70%) and larger P erosion increase with P in the similar way when V=10% (and smaller P), but as P gets smaller in the drier phase of the oscillation erosion does not drop as much as the less dense (and drier) simulation. Why the model gives this asymmetric response in E for given P oscillations (for drier and wetter P) for V=0.7 needs to be explained by the authors, as this is a very interesting result. Also why does the simulation with V=0.7 can double its erosion similar to V=0.1.? The mean erosion is higher than the case with V=10%. 10 cycles were plotted, given a total model year of 1M. I wonder why only the negative changes in P was dampened by vegetation in the denser V case

but not the wetter cycles? This probably has to do with the way slopes adjusts under the two climate regimes. In both simulations you have a steady-state landscape as initial condition, and when P grows, is the erosion threshold surpassed in all locations resulting in a very similar erosion magnitude?. How long would the landscape need to attain a dynamic equilibrium under cyclic climate?

Fig 15.Vand P > or < V and P is not very informative.

---

## Short Comment (SC2) · 20 Apr 2018

This manuscript has a lot of value in terms of exploring the implications of some of our current best guesses with respect to how vegetation cover and precipitation modulate erosion rates at the Earth surface and influence topographic evolution over long timescales. While the investigation of the individual influences of precipitation and vegetation cover provide insights into the role of each, the dynamic behavior that arises through the combined modeling of both comprises a testable set of results, which could lead the way to improved equations in the case of clear mismatches. People in our community with a focus on details of individual processes and individual types of vege-

tation cover may find the generalization necessary for such models running over such long timespans to be an oversimplification, but I agree with the approach of first testing whether or not this simple approach can explain first-order observations with respect to erosion rates and landscape morphology.

Along these lines, I think the authors could expand their discussion somewhat to better emphasize how those of us from the field-data side might help to test these results. Testing the results of the model with respect to the morphology of the Andes is useful, but also has some broad limitations. For example, it's difficult to know if the topography that we see today is fully adjusted to forcing conditions such that it reflects the predicted influence of precipitation and vegetation cover, or if there could be a persisting transient response to tectonic activity. Calibrating the model to the conditions in the Andes could be problematic if the landscape is currently in a transient adjustment state. I wouldn't suggest that this issue rules out the possibility of comparing model predictions with topography, but it does point to a reason why there might be mismatches.

Another possibility for comparison would be to consider datasets that have recorded temporal variations in erosion rates in response to changes in precipitation and vegetation cover, particularly over the precipitation and vegetation cover ranges that the authors suggest drive the biggest changes. Field data that support the model results (particularly Figure 17) would be a strong argument in favor of the equations used. Unfortunately, those datasets are somewhat rare; the only ones I am aware of that report both vegetation cover and precipitation changes include Marshall et al. (2015, Science Advances) and Garcin et al., 2017 (EPSL), although Marshall et al. was mostly focused on frost-cracking, which would be less relevant for the model presented here. While I'm a co-author of the second and don't want to insist that the authors consider that study, I think it could be valuable, as we found quite strong variations in 10Be derived erosion rates reflecting the onset and persisting influence of the African Humid Period. An in-depth comparison of your model results to those data would clearly be out of the scope of the manuscript, but a qualitative comparison could be useful.

One aspect of the modeling results that surprised me is that for the 10% vegetation cover case, increasing precipitation leads to a decrease in erosion rates in the model. I find this somewhat counter-intuitive, and contrasts in some ways with what we see in East Africa, where the onset of wetter conditions leads to a brief spike in erosion rates, however the rates rapidly decreased once denser vegetation cover (trees, compared to a mix of grasses and trees earlier) established itself. So why the difference with your results? I'm guessing there's not much of a time lag in the model between increased precipitation and increased vegetation cover... if there were, perhaps the model would show a response more similar to what we see in our data.

I'm unsure whether short short-term responses are important for landscape evolution models running on million-year timespans. But even if it's brief, a spike in erosion rates could have a reasonably good change of being preserved in stratigraphy, so there may be a good chance of recovering such responses with field data. Along the same lines, I wonder whether or not it would be possible to record (with field data) the high-frequency variations that the model shows during the declining-precipitation phase (again for the 10% veg cover case). Do you think we could we resolve a sudden, brief decrease in erosion rates with 10Be data? Two reasons why it could be tricky is that the thickness of any stratigraphic layer would be limited, and also at lower erosion rates, the erosion rates are integrated over longer timescales. This point may be worth discussing.

Some additional minor points:

Section 4.1, lines 248, 255, 266: It would be helpful to use the term "spatially variable" in this section rather than "variable" alone, to make it clear that you are not focusing on temporal transients.

Why isn't section 5.4 in the results section (section 4)?

Figure 16 is hard to interpret without Figure 17; it would be easier if the forcing (change in precip) were shown in Figure 16 or at least explained in the caption. Are changes in vegetation cover modeled or prescribed? I'm embarrassed that I was reading too

quickly to discern that detail, but in any case, it would be helpful to have that information more prominent.

---

## Referee Comment (RC2) · Anonymous Referee #2 · 23 Apr 2018

The paper uses an established landscape evolution model (Landlab) to evaluate the effects of precipitation and vegetation cover change separately and combined on a myr time scale. It is timely and addresses the important question in earth science if vegetation is a main driver of denudation, and hence fits well into ESurf. Thank you for letting me review this manuscript; I am not a landscape evolution modeller, and hence it was a challenge for me, and I have to leave more technical comments to the experts. I find the topic and results fascinating though. My perspective is more process-orientated, and this is also where my criticism, but also fascination originates in. Sorry for the delay. I find the paper overall well written, but it is too thick in times. The discussion suffers from being too long at the one side, but could gain a lot from a comprehensive figure that summarizes the outcomes conceptually. Please consider that not all people who are interested in this topic have experience in landscape evolution models, and have potentially never seen the outputs of Landlab before. This is also especially important with regards to the Figure captions, which are often not specific enough. Please also add something to the title that clarifies the type of study, e.g. the time period considered or/and that it is a landscape evolution model study. I have two main criticisms that made the paper more cumbersome to understand; the first regards the origin of the vegetation cover and the oscillation part of the paper, it is not clear on which base you chose these assumptions. The second is that the title covers a large topic; however the interpretation of your output is quite limited and stays very close to the model output. It doesn't include literature or discussion points from studies outside of the landscape evolution world, e.g. the effects of knickpoint retreat, or an interpretation from the process-domain, e.g. denudation rates on deforested catchments without vegetation cover (rates summarized e.g. in Montgomery, 2007). From my perspective, there are two ways to resolve this, either you claim a larger importance and add e.g. an overall conceptual figure and include literature from other fields, or you modify the title and narrow it to the landscape evolution world, which is what I would opt for. I think this would also reduce the weight of earlier criticism of the paper which I understand where it comes from. The fact that you apply an average vegetation cover, hillslope denuation and river incision is represented in the same equation, and that there is no representation of groundwater in the model justifies the question what the significance of the study is, and I suggest to try to do a better job in clarifying this. In parts it sounds like the reason for this paper is to develop the model setup for the following papers, which doesn't really help to assess how your paper advances science. Generally, I find the mix between a setup of non-natural conditions (e.g. precipitation without vegetation change) in combination with the "loose" tuning to the Chilean catchments problematic. If you would like to investigate the effects of both, precip and veg independently, then why not use a catchment that has equally distributed aspects and slopes, so that you can make more comprehensive interpretation of how catchment topography controls

the flux? Please try to avoid to mention that you will model the evolution of the catchments in more detail later, this leaves the taste of salami-slicing. The study should stand for itself. The same is also true regarding the companion paper. I miss more references in the method section, so that it is clear what of the approach is "best practice", and which you developed yourself or used for the first time. Figure 17: Please explain more in detail where these result come from; e.g. the dotted line in b should look more like in a in the grey field?

---

## Editor Comment (EC1) · R. Hodge (Editor) · 10 May 2018

I am happy that the authors should proceed on the basis of the comments and reviews received so far. The reviews are admittedly mixed, but on the whole the reviewers feel that there is merit in this work. When addressing the the points that the reviewers identify, the authors should consider how they present and justify the model parameter selection, and whether any model calibration is possible or appropriate. There are also some suggestions for how the model results could be broadly compared with examples from the literature. As identified by a couple of reviewers, a conceptual model might help to summarise the model findings. Finally, I would also encourage the authors to

consider whether there are other parts of the paper that could be made more concise.

**ESurfD**

---

## Author Response (AR1)

**RESPONSE TO REVIEWS - ESurf Manuscript**

**Effect of changing vegetation on denudation (part 2): Landscape response to transient climate and vegetation cover**

By: Schmid et al.

Responses in blue, original comment in black.

**Response to Associate Editor: Rebecca Hodge (Univ. Durham)**

I am happy that the authors should proceed on the basis of the comments and reviews received so far. The reviews are admittedly mixed, but on the whole the reviewers feel that there is merit in this work. When addressing the the points that the reviewers identify, the authors should consider how they present and justify the model parameter selection, and whether any model calibration is possible or appropriate. There are also some suggestions for how the model results could be broadly compared with examples from the literature. As identified by a couple of reviewers, a conceptual model might help to summarise the model findings. Finally, I would also encourage the authors to consider whether there are other parts of the paper that could be made more concise.

**Dear Prof. Hodge:**

At first we like to thank the referee's and the editor for the very useful comments on our manuscript which we hope will improve the quality of the presented research and make it more useful for other interested readers. We also would like to thank Taylor Schildgen and Erkan Istanbulluoglu who used the open discussion to bring in their valuable expertise to the manuscript in the form of short comments, which were also very helpful.

We hope that the revised manuscript we present here helps with clarification of the referee's criticism and fits the high standard of esurf.

The most extensive changes to our manuscript are in the Background and the Methods section to address the points of criticism that referee's brought up about the readability and clarity of our model-approach and parameter selection. Discussion section 5.6 (Model Caveats) has also be expanded to address reviewers comments. We also added a new subsection 2.1 which we hope will explain the general approach we used for conducting our model experiments. We also tried to create clearer reference to the applicability of our results to a field-application by adding new discussion section (section 5.5). Since both, Taylor Schildgen and Erkan İstanbulluoglu, as well as Referee 2 brought up the point that a conceptual model or at least a figure describing our results more conceptually would help with understanding the results, we hope to find a good middle-ground with the new figure 18. Since most referee's commented that the paper is too long in general, we reduced text throughout the manuscript to make it more easy to read and to reduce repetition, but due to additional paragraphs and sections we needed to add to address the other valuable points of the referees, the paper did not become shorter at the end of the review process. However, for the longer-format version of ESurf we prefer to have a thorough manuscript that provides a valuable reference for interested readers rather than a short manuscript that is missing sufficient details for a thorough understanding.

Attached in this document are our line-by-line answers to the referees comments and we also decided to answer the short comments, because we found them very helpful and wanted to acknowledge that these scientists took the time to review

a paper for which they were not the designated referees which is a nice effort and in the spirit the open-discussion format of EGU journals.

At the end of the document the revised manuscript with tracked-changes is attached.

Please let us know if you have any questions. Sincerely, Manuel Schmid and Todd Ehlers (corresponding author) on behalf of all authors.

**Response to RC1**

Responses in blue

Summary: The authors attempt to link changes in landscape form to changes in climate and vegetation. Although this is a worthy topic to explore, it would be a challenge for even a more complicated model because we simply don't know enough about the processes and feedbacks involved. Considering that this model greatly oversimplifies what we actually do know, its contribution to our understanding is not obvious. Those familiar with these processes will view the results with skepticism, and those who aren't may believe the results without fully appreciating all of the short-cuts and assumptions baked into the governing equations. I know how much work goes into modeling exercises like this so I always try to be open-minded when reviewing these types of manuscripts but, in this case, I cannot recommend publication.

1) The reviewer raises an interesting point, related to different philosophies to modeling of landscape evolution. We respectfully disagree with the reviewer's assertion that the community simply doesn't know enough about the processes involved to approach the problem. Science progresses by a stepwise confrontation of what we don't know. The approach followed in our study is to identify emergent behaviour based on a simple set of assumptions. As more is known (from much needed field studies) about how to better parameterize models then of course improvements can (and hopefully will) be made on our approach. As highlighted below, our manuscript builds upon approaches already published in the literature. We also respectfully disagree with the reviewer that a more complicated model is needed at this stage. Inclusion of a more complicated model at this time would only result in including poorly constrained parameters (particularly over the long-timescales we investigate). In the following response to this review we expand upon the difference of opinions in how to approach this problem, and also address these reviewers points, many of which we agree with and can readily implement into a revised manuscript.

Based on the comments below we believe the reviewer comes from the perspective of short-term observations of detailed eco-hydraulic processes. While this field of study is definitely useful, that is not the scope of this study, where we are interested in long-term (millenial to million year timescale effects of vegetation and climate on topography). There simply is no way to include short-timescale ecohydraulic modeling approaches into a study that is simulating million-year timescale processes. This is because a) computationally CFD and 3D (e.g. hydrogeosphere) type modeling approaches can not be conducted over these timescales, and b) the data inputs needed to constrain models over long time scales do

not exist. Thus, a sensitivity study such as we present that seeks to identify emergent behavior between vegetation-climatesurface process interactions is the first place to start.

We refer the reviewer to a publication of Dietrich et al. 2003, which nicely summarizes different perspectives of how landscape evolution models are meaningful to answering scientific questions. More specifically we follow a approach Dietrich described as 'essential realism' whereby:

"This condition combined with non-linear, threshold dependent erosion processes leads to a significant component of indeterminacy in the evolving topography. Therefore, it is unrealistic to expect to predict the exact topography of a landscape at any particular time, including the present. Instead the gross trends, the quantitative relationships, such as illustrated in Section 2 and the references cited therein, are the features landscape evolution models can realistically hope to explain."

Thus, by the very nature of how this study is set up we are interested in an emergent behaviour in vegetation/climate interactions over geologic timescales. Thus, while we appreciate the reviewers perspective, she/he is looking at this problem from a very different perspective, and disregarding how an entire sub-community of geoscientists approaches modeling of Earth surface processes.

We note that in the reviewers comments below she/he has issues with the governing equations and modelling approach used, we note that we follow the published approach of Istanbulluoglu and Bras, 2005 and highlight that this study lacks the detailed consideration of processes the reviewer would like, yet has had a highly significant impact in the field of long-term landscape evolution modeling (98 citations). Furthermore, the general approach of applying landscape evolution models over long timescales has provided many valuable insights into different geomorphic and geologic problems (Howard 1994, Tucker and Slingerland 1997, Whipple and Tucker 1999, Collins and Bras 2004, Jeffery et al. 2013, Yanites et al. 2017, and many more ....)

Nevertheless, we realize many other readers of this journal, will share a similar perspective of this reviewer, so in the revised version of the text, we will highlight these contrasting perspectives in the start of the background section with a new subsection (landscape evolution modeling and the applicability of these results). We hope that these changes and also the other changes outlined below will reach a happy middle-ground with this reviewer, where the (essential-realism) approach of Dietrich et al. is recognized as a valid way of understanding emergent behaviour in systems for processes that are to complicated to simulate over geologic timescales.

My comments are mainly focused on the governing equations. Because I believe them to be either unsupported or flawed, I don't address the results in detail. If the governing equations of a model are not honoring reality in some fundamental way, then its output will be unreliable.

2) We are unable to respond to this comment without details from them. We respond to the detailed examples given below. We agree with the reviewer that well-calibrated field-studies of geomorphic transport laws are lacking in the literature, and we have based our analysis on what little information is published.

It didn't seem like the authors tried to determine whether the model was working correctly. Thanks to 10Be, we have erosion rates for many watersheds around the planet, including catchments that are similar to the ones modeled here. Before we can believe that the model works, the authors ought to run it on one with published erosion rates. This would have been an important first step before embarking on the rest of the project.

3) We apologize for any confusion at this point, the manuscript currently states that the goal is not providing a calibrated study of the Chilean Coastal Cordillera (see Introduction, last paragraph and Results section 4.). This is because catchment average 10Be denudation rates have not been published for this areas, nor are river sediment-flux data available. The rock uplift rates used, are based on the long-term averages from thermochronology, as cited in section 3.2. However, we have to disagree with the reviewer on two grounds that 10Be measurements would be useful here: (1) even if 10Be measurements were available, they would not be directly comparable to this study because the integration timescale of in-situ produced 10Be will be 10's to 100's of years such that average rates over these timescales would be produced (e.g. see Schaller and Ehlers, 2006 EPSL), and comparison to predicted erosion rates vs. a higher fidelity time scale would not be meaningful. (2) We again emphasize (see starting response to reviewer) that the goal of landscape evolution modeling is not to reproduce reality. This simply isn't possible given how little we know about the relevant processes over millennial timescales. Sensitivity studies based on what is known (as done in this study) are the first step forward.

Given that our entire model setup and approach is a sensitivity analysis to poorly understood processes acting in the Chilean Coastal Cordillera, we have evaluated model performance using first order topographic metrics (e.g. relief, slope, and Ksn) for the application area. Comparing model predictions to these first order attributes of topography avoids over interpreting the model results. A more detailed comparison than this, would require data that doesn't exist. Thus, we've framed the interpreted model output and modeling approach around first order topographic metrics for these area. This is a conservative approach to interpreting climate and vegetation effects on topography.

To address this concern raised by the reviewer, we will add additional text in section 3.1 to emphasize more strongly that this is not a calibrated study of the Chilean Coastal Cordillera.

This model is driven by, essentially, two governing equations. The first describes soil creep via linear diffusion. The authors use a formulation, proposed in a paper from 2005, that links the diffusivity to vegetation density that was based on little data and only accounts for physical processes (eg, rainsplash) and ignores bioturbation. Are there field observations to support that physical processes dominate soil creep at their field site? Importantly, there is no support for the nonlinear equation that relates diffusivity to vegetation density.

4.) We thank the reviewer for highlighting that clarification is needed on this point. The approached used in our study was first published in models by Istanbugulough and Bras, 2005, and Collins et al. 2004. The field/laboratory investigations supporting a negative exponential relationship for diffusivity Kd as a function of vegetation cover comes from Alberts et al., 1995, Dunne, 1996 and Dunne et al. 2010. Thus, there is a history of peer-reviewed articles supporting our approach.

To address this issue we will modify the manuscript to add the above references.

Over a narrow range of precip, Ben-Asher et al (2017) found a linear inverse relationship between diffusivity and precipitation which, when combined with the present study's relationship between veg and precip might yield something like the negative exponential equation adopted here but the authors have not demonstrated that. Moreover, a paper that examined diffusivity across a wider range of precip (Hurst et al, 2013) found the relationship to be weakly positive – which runs counter to what the authors have assumed. There is little support, then, for the way the soil creep equation has been parameterised.

5.) Thank you for bringing these references to our attention. The Ben-Asher study is interesting, although we can not apply it here because they link diffusivity to precipitation, not vegetation change. The Hurst et al. paper is unfortunately not usable for our purposes. First of all because experimental design of the study looks at a diverse range of lithologies and finds essentially no correlation between precipitation and diffusivity (sediment transport coefficient). The Hurst paper clearly states in the Fig. 1 caption that the weakly positive relationship has a R2 =0.27 (n=24) for a linear regression of only a subset of the data. For the reviewer to conclude there is 'little support' for our approach based on a R2 of 0.27, from a study that doesn't consider vegetation differences does not seem to be a correct application of the Hurst study to ours. Furthermore, in the Hurst study, the authors do conclude that they find two different values for basins with the same vegetation density but those catchments were situated in a different lithologic regime and so the authors themself conclude that this difference is probably due to soil properties emerging from different underlying lithologies.

The second governing equation describes erosion by flowing water. Before describing my main concerns, I should point out that this section (lines 193-215) was difficult to follow and was missing some critical details. For example, variables appear without explanation or description and the relationship between eqn 5 and the others that follow was unclear. Also, there was no explanation of how rainfall is applied (eg, storm frequency and magnitude), how runoff is generated, or how runoff generation is affected by vegetation density (eg, via interception). The point about storm frequency and magnitude is especially important because changes in climate will affect the distribution of both of these but not necessarily in a uniform way. Given that, here are my main comments regarding the way that erosion by flowing water is treated in the model.

6.) To address this we will modify the text to clarify how eqn 5 links to the other equations. Thank you for noticing this. We will also clarify some problems with parameter-descriptions in the equations and apologize for the inconvenience and not catching this in the first place.

Concerning the surface water hydrology - we will add text in Section 3.2 to explain the approach used. Our approach is similar to other long-term landscape evolution studies and we provide references (Croisssant and Braun, 2014, Jeffery et al. 2014, Yanites et al. 2017). The reason for using mean annual precipitation in many long-term landscape evolution modeling studies is (a) information about paleo (and even modern in many cases) precipitation duration, intensity, and interval are not known and inclusion of an unconstrained parameter in a model is counterproductive for a sensitivity study, and (b) the model simulation time step can be significantly larger (e.g. 100 years in our case) when mean annual precipitation is used - thereby allowing millions of years to be considered. However, including storm events increases simulation time dramatically by requiring hourly time steps and can make studies like this intractable.

To address this issue we will address this comment in the expanded the text in the model caveates section (Discussion section 5.5) to address these points.

1) Linking the roughness coefficient to the vegetation in the way that was done here ignores the fact that the effect of vegetation goes beyond a simple measure of 'vegetation density.' For example, imagine two landscapes with 70% vegetation density: one is covered by shrubs such that the ground surface between each plant is essentially bare while the other is covered by grasslands. These two landscapes, despite having the same vegetation density, will have different manning's n values on the hillslopes. Since we know that vegetation community changes with climate, the model's attempt to scale manning's n on the basis of vegetation density is not realistic. Indeed, I looked at the field sites via Google Earth and it was clear that vegetation community does change as a function of precip in those regions. I can easily imagine situations where manning's n actually increases with a decrease in vegetation density, the opposite of what is assumed here.

7.) The reviewers comment states that the metric of "vegetation cover" that we used in our study oversimplifies the interaction between different plant communities and mass transport and erosion. While we want to acknowledge that this is a very good point which certainly holds true, we also want to defend our decision in using vegetation cover. The available satellite data which gives one the possibility to make a spatial distinction between different types of vegetation has a resolution of 500m, which would resemble 5 grid cells in our model domain and represents a integration over 11 years, which makes it hard to extrapolate these data to a distinct vegetation-community for longer timescales. We argue that, our approach of applying a very simple transient forcing which resembles a change in "simple vegetation-density" is probably not resembling reality, it would still be much harder to get a realistic transient time series of shifts in plant functional types for changing climatic conditions. This is however part of ongoing research to incorporate a fully functioning dynamic vegetation model into this landscape evolution exercise.

To address this comment - we will modify the methods section and section 5.5 of the text to mention this caveat and to highlight to readers that this is a simplification that can be hopefully improved up with additional data sets and calibrated erosional laws in future studies.

2) Given the comments above, both landscapes will also have different critical shear stresses. For example, soil with shallow grass roots will be more difficult to erode than the bare soil between shrubs. It doesn't appear that the model takes this into account.

8.)While the reviewer is certainly right that the shear-stresses would certainly differ, in this study the goal was to have as few free parameters between simulations for different study-sites as possible, therefore we decided to focus on the effect of vegetation cover to the river erodibility factor K. Given that condition, we argue that choosing a common critical shear-stress is reasonable because of the uniform substrate lithology used throughout the entire simulation duration.

To address this comment - we will modify the methods section and section 5.5 of the text to mention this caveat and to highlight to readers that this is a simplification used.

3) It appears that the model doesn't distinguish between overland flow on hillslopes and river flow. If so, the authors are assuming that a source of roughness on the hillslopes – the vegetation – is also contributing to roughness in the rivers. For example, if the authors envision shrubs growing on their hillslopes then they must also be growing in the rivers. Again, I don't see how this is realistic. Moreover, the type of vegetation really matters here with respect to flow depth. Short grasses would have a greater manning's n with low flow depths (ie, overland flow) than with deeper flows (ie, rivers). Conversely, shrubs would have a lower manning's n with overland flow than with river flow.

9.) The reviewer is correct that the stream-power model widely used in the literature and modeling studies does not distinguish between different regimes for surface flow vs. stream-flow. The dominant effect of diffusive hillslope processes over advective stream-flow processes is assumed to be regulated by the critical shear stress and the ratio of diffusive material flux vs advective material flux which is a commonly used approach in large-scale landscape evolution modeling (e.g. see references cited above). The decision to keep vegetation cover spatially uniform over the modeldomain comes from the poor constraints about how effective channelized flow in rivers actually removes the superimposed vegetation cover. Previous studies linked the removal of vegetation to bed-shear stress within a riverchannel but the relationship on how effective this process works is still not well understood. There are two processes to consider here: it still is very unclear how different types of vegetation are actually able to withstand surface shear-stress because of bending of branches and stems and, if they are removed, how fast they will grow back. The second process is the adjustment of the ecosystem to stream-flow by shifting the vegetation in or near a river channel to plant functional types that are more accustomed to these positions which would certainly lead to a shift in vegetation type but it is unclear how this will act on the vegetation cover metric. Also, while the reviewers comment will certainly hold true for larger basins with larger rivers and more diverse vegetation types, from field-observations that we made within the focus areas, it emerges that most of the stream-bed is actually made up of a dense root-network and vegetation cover very similar to the surrounding hillslopes because of the small catchment sizes and 100 m model resolution.

We will address this reviewer's concern by modifying the manuscript in Section 5.5 (Model caveats and restrictions) to mention these complications in that they are not represented in the modeling approach because a means for scaling these processes to long time scales is not known.

4) There was no explanation of how the critical shear stress was calculated. Presumably some assumptions were made regarding bed and hillslope material but these were not described. Does the model keep track of the evolving particle sizes as climate changes? More vigorous runoff will coarsen the river beds and hillslope surfaces but I didn't get the sense that this was incorporated into the model. Also, a lower vegetation density will expose the ground surface to raindrop impacts that will mobilize finer material more readily. Was this accounted for? Again, it didn't seem like it.

10.) As the reviewer correctly mentioned the critical shear-stress was chosen in accordance to values presented in other studies for granitic underlying material. We want to point out that this model was set-up as a 1-layer detachment limited case, following the Fastscape Algorithm developed by Braun and Willet, 2013, which resembles a bedrock dominated landscape where the ability to transport eroded material out of the system is the main driver for the evolution of river

catchments. This detachment-limited problem formulation is, even though the general transport formulations are still discussed in recent geomorphic literature (e.g Davy 2009, Pelletier 2011) used extensively for a variety of different landscape evolution models and is believed to produce realistic results for headwater channels (Howard, 1994). We agree that landscape evolution models would benefit greatly from more effective algorithms which would incorporate more hydraulic parameters but studies have shown that even more complicated models which incorporate a higher-level of water-/bed- interactions fail to produce a clearly better prediction of channel morphology (Turowski et al. 2007). Therefore we agreed to use a simple detachment limited formulation for the landscape evolution model which lacks the ability to track evolving particle sizes of river-bed and hillslopes and to incorporate the effects of coarsening/fining of bedstructure back to critical shear-stress.

To address this comment, we will modify the manuscript in the model setup section to add the above references for how the shear stress is calculated.

One of the model's limitations are well-illustrated by Figure 9. It predicts long-term erosion rates on the order of about 0.2 mm/y. This is on the high end of known soil production rates; I would venture to guess that soil production rates at the sites in Chile are quite a bit lower given the dry conditions (0.2 mm/y is what you get in weak-ish bedrock in the Oregon Coast Range where its wet and has lots of trees doing physical weathering). This means that, at these high erosion rates, the landscape would run out of soil yet the model seems to assume an inexhaustible supply of erodible material. In the real world, the loss of soil would have important consequences for runoff processes and the ability of plants to grow but the model seems blind to these.

11.) The reviewers comment links to the answer we gave above about the detachment-limited setup of our model which assumes a bedrock-dominated landscape and neglects the effects of different soil covers on erosional processes. The 0.2mm/yr long-term erosion rates are a product of the conservation of mass approach within our model domains, which experience a uniform tectonic uplift of 0.2mm/yr. This value is supported by thermochronology studies done in Coastal Cordillera catchments north of the location, and we cite the reference for this value used in the paper. We acknowledge that these regions may have a different uplift history than our focus areas in the Coastal Cordillera, but provide the best dataset for uplift estimation in this region, but there is currently no other observational studies published that constrain this value better. We hope that other studies, done within the Earthshape project, specifically done to determine weathering rates and catchment-wide erosion rates based on 10Be will help to better constrain these input parameters. Finally, we agree with the reviewer that approach assumes a temporally and spatially constant material is being eroded, and the transition from soil mantled to bedrock lithologies would potentially introduce a different response. However, we can respond that (1) data are not available to provide a believable prediction of soil production rates, (2) introducing an additional processes into the modeling (e.g. the transition from soil/regolith to bedrock mantled landscapes) could be a study on it's own, and (3) there is currently no reason to believe apriori that the landscapes ever were stripped on their soil/regolith such that the erosivity would would vary. Rather than introduce these uncertainties into our analysis, we follow the approach of many other modeling studies working on these timescales and assume the substrate material properties remain constant through time.

Finally, there was no attempt to provide any error estimates in the predictions. I understand that this is not common practice with landscape evolution models but it should be and can be done (see papers by Tom Dunne on stochastic modeling). For example, the model makes certain predictions about how erosion rates may vary over time after changes in vegetation (Figure 9). Given all the potential uncertainties embedded in the governing equations and how they were parameterized, how confident are the authors that a predicted erosion rate of 0.2 mm/y is statistically different from a predicted C4 erosion rate of 0.4 mm/y? I'm skeptical that this model can predict annual erosion rates accurately to one tenth of a millimeter. My concern is that models like this one, while perhaps useful for demonstrating basic principles to students, are not well-suited for answering important scientific questions, especially when they haven't been tested under the relevant circumstances.

12.) We believe that the reviewers comments are oriented towards a smaller-scale, shorter timescale analysis. As she/he pointed out already, large-scale landscape evolution models are not always fit for implementing these analysis. While we see the merits of knowing the uncertainties in the predicted values, this requires known uncertainties in the observations / input parameters to implement a stochastic approach. For example in this study, it would be hard to define an uncertainty e.g for the change of mean annual precipitation with vegetation cover. We could implement a range of these changes, extracted from other regions on Earth but this would not be an error estimate but solemnly the product of other regional boundary conditions in these regions. Furthermore these approaches are not common in long-term landscape evolution studies because the emphasis (as we started our response to the review) lies in the exploration of emergent behaviour due to transient forcings and not to reproduce reality. We acknowledge that a landscape evolution model which would be able to reproduce exact replications of landscapes and inherent fluxes would be best for the scientific community, but, still due to the problem of some poorly understood processes and constrained variables, we think that there still lies value in focusing on simple models and analyse general behaviour to gain a better understanding of possible underlying, large-scale processes.

Finally, the reviewer's statement that this study is "...perhaps useful for demonstrating basic principles to students, are not well-suited for answering important scientific question,..." is unconstructive. We are not aware of any other study in the literature that demonstrates the counter intuitive and non-linear responses demonstrated in this study. We would find merit in this comment if the reviewer evaluated the results presented and discussion (text and figures) and highlighted how this is already a well-known result.

**Response to RC2**

Responses in blue

The paper uses an established landscape evolution model (Landlab) to evaluate the effects of precipitation and vegetation cover change separately and combined on a myr time scale. It is timely and addresses the important question in earth science if vegetation is a main driver of denudation, and hence fits well into ESurf. Thank you for letting me review this manuscript; I am not a landscape evolution modeller, and hence it was a challenge for me, and I have to leave more technical comments to the experts. I find the topic and results fascinating though. My perspective is more process-oriented, and this is also where my criticism, but also fascination originates in. Sorry for the delay. I find the paper overall well written, but it is too thick in times.

The discussion suffers from being too long at the one side, but could gain a lot from a comprehensive figure that summarizes the outcomes conceptually. Please consider that not all people who are interested in this topic have experience in landscape evolution models, and have potentially never seen the outputs of Landlab before.

1. Thank you for this interesting and thorough review which helps a lot with advancing the scope of the study. Actually the shown figures are not landlab-specific output but timeseries of topographic metrics. We acknowledge that this could be solved by also addressing your next point of criticism, with more specific figure captions.

To address the reviewers point we have added a figure to visualize the concepts behind the model more clearly.

Concerning the text being too thick in parts: We have done our best to shorten the manuscript where possible, but the large number of reviews received for this paper (4 total) have required text additions to clarify points so the overall length of the manuscript. ESurf is a long format journal article venue and we chose it specifically to have sufficient space to explain concepts as needed, while keeping the text as concise as possible.

**This is also especially important with regards to the Figure captions, which are often not specific enough.**

2. See the answer to paragraph above.

To address the reviewers point we will rework and expand the figure captions to make it more clear to readers to understand the underlying data.

Please also add something to the title that clarifies the type of study, e.g. the time period considered or/and that it is a landscape evolution model study.

**3. Thanks for this thoughtful advice**

To address the reviewers point we will try to make the title more specific about the type of study that was conducted

I have two main criticisms that made the paper more cumbersome to understand; the first regards the origin of the vegetation cover and the oscillation part of the paper, it is not clear on which base you chose these assumptions.

4. Thanks for bringing this to our attention. The basis for choosing the vegetation cover values was data from the modismission for catchments situated around the Earthshape focus areas. The oscillating time series was chosen as an approximation of 100kyr milankovitch cycles which has been identified as one of the main frequencies in Earths climate cycles (Broecker & van Donk, 1970, Muller and MacDonald, 1997). This frequency was chosen because on the long-timescale our simulations were conducted, the 100kyr cycle is the most stable cycle with the highest periodicity induced by planetary motion.

To address the reviewers points we will add/modify text in the manuscript in section 3.2 and potentially figure 5 to make this more clear to readers

The second is that the title covers a large topic; however the interpretation of your output is quite limited and stays very close to the model output. It doesn't include literature or discussion points from studies outside of the landscape evolution world, e.g. the effects of knickpoint retreat, or an interpretation from the process-domain, e.g. denudation rates on deforested catchments without vegetation cover (rates summarized e.g. in Montgomery, 2007).

From my perspective, there are two ways to resolve this, either you claim a larger importance and add e.g. an overall conceptual figure and include literature from other fields, or you modify the title and narrow it to the landscape evolution world, which is what I would opt for.

5. That is a good and well-reflected point, thank you! We'll try to get a better representation on the scope of this study by adjusting the title and adding more explanation in the background part of the paper. We acknowledge that a model-setup like the one we used with is probably not suitable for answering questions of soil-erosion through agriculture because of the 1-layer setup assuming detachment-limited conditions that are thought to prevail in bedrock dominated basins. The Montgomery study you brought to our attention, while being a very interesting and helpful study aims to quantify exclusively matters of soil-erosion rates on agricultural landforms which is not the real scope of this study. To address the reviewers point we will make it clearer in the title which field of study this paper aims to address

I think this would also reduce the weight of earlier criticism of the paper which I understand where it comes from. The fact that you apply an average vegetation cover, hillslope denudation and river incision is represented in the same equation, and that there is no representation of groundwater in the model justifies the question what the significance of the study is, and I suggest to try to do a better job in clarifying this.

6. Thanks for this point of criticism. We want to clarify that we do not apply an average hillslope denudation and river incision to our model, but that those values are results of this as the different differential equations for hillslope diffusion and fluvial erosion are solved over each node. We do however apply a spatial uniform vegetation cover on our focus areas. We acknowledge that this is a simplification of reality but due to the conception of our model which focuses on quantifying the first-order behaviour of topographic metrics for different settings of initial vegetation cover, we would argue that this is a reasonable approach.

To address the reviewers point we will add text in section 3.2 to explain the justification behind this approach better.

In parts it sounds like the reason for this paper is to develop the model setup for the following papers, which doesn't really help to assess how your paper advances science. Generally, I find the mix between a setup of non-natural conditions (e.g. precipitation without vegetation change) in combination with the "loose" tuning to the Chilean catchments problematic.

7. We thank the reviewer to point out that we need to clarify the general model setup and why we choose these non-realistic conditions of varying only precipitation or only vegetation cover in combination with metrics specific for the Chilean catchments. The general idea behind this was to delineate the separate effects of a coupled system (vegetation / precipitation) on landscape through specifically designed model experiments. Another way of doing this/ thinking about this would be to conduct a large-scale flume experiment with constant imposed rainfall and different values of vegetation cover and vice versa but numerical modeling brings us the opportunity to explore these relationships and explore the implications of existing physical relationships proposed in literature.

To address the reviewers point we will clarify why we have chosen these specific transient forcings in section 3.2

If you would like to investigate the effects of both, precip and veg independently, then why not use a catchment that has equally distributed aspects and slopes, so that you can make more comprehensive interpretation of how catchment topography controls the flux?

8. Thanks for this suggestion. I think this is aimed at the comparison of our model results to the topographic data that was extracted from the Earthshape focus basins. We would like to point out that in addition to basins that share equally distributed aspects and slopes, another important factor would be to that the basins also need to show same underlying lithology and we would need additional ground-truthing that the detachment limited case we apply in the model-setup holds true for the observed catchments. The Earthshape catchments were also chosen as part of the dfg-funded priority research project with the aim to produce inter-comparable data between different projects. We are unaware of a better location to conduct such a comparison that contains a similar tectonic setting, lithology, and large climate and ecological gradient.

Please try to avoid to mention that you will model the evolution of the catchments in more detail later, this leaves the taste of salami-slicing. The study should stand for itself. The same is also true regarding the companion paper.

9. Thank you for this feedback. The reason why we mentioned the ongoing work of modeling these catchments is that we are in the process of developing a coupled model-setup between a state-of-art dynamic vegetation model and this surface process model, but we see your point and have modified the manuscript.

To address the reviewers points we will try to clarify that this study is not dependent on the results of these future models but can be more thought of as setting the interpretation-framework for future studies in this direction.

I miss more references in the method section, so that it is clear what of the approach is "best practice", and which you developed yourself or used for the first time.

Thank you for bringing this to our attention.
 To address the reviewers point we will add more references explaining the basic model setup and the underlying equations.

Figure 17: Please explain more in detail where these result come from; e.g. the dotted line in b should look more like in a in the grey field?

11. To address the reviewers point we will elaborate more about the interpretation of these results in the figure caption and the discussion section of the paper.

**Response to SC1**

Responses in blue

This is a very interesting paper. I have some suggestions and general comments for improvement and clarity of the theory used. The length of the paper can be reduced. I felt some descriptions were repetitive under various subheadings which can also be reduced. It follows a fairly standard writing style that described methods, results, discussion etc. but in each of these certain methods are repetitive. For example 4.1, 4.2. are part of the results section but there are paragraphs that still tell what was plotted w/o giving results.

1. First of all we would like to thank you for this constructive and precise review, even though you were not appointed an official referee. We acknowledge that it is a long paper. The repetitive parts you are referring to, were first incorporated into the paper to make it easier for readers to start reading at a specific paragraph of the results section but we agree that its redundant information and makes it harder to read through the whole paper, so we tried to cut the unnecessary information in the relevant paragraphs

In the discussion of the model results quantitative details were presented. Given the model was not calibrated for the study watersheds, I wonder if those details matter or realistic at all. This study could lead to more qualitative results that can be summarized/discussed in a conceptual model. But in order to do that the number of simulations done may not be sufficient.

2. That is a good point. We agreed on giving the quantitative results because, while the model was not exactly calibrated to the different areas, the result of the model runs were verified by comparing the main trend of topographic metrics produced by the model for the different areas with the metrics obtained by DEM-analysis. We added a better explanation of this in 2.1

My main concern is that the results presented here for constant and transient changes individually P and V and combined, is only one potential outcome of a wider range of responses. The paper generally does not ask the "why" question in presenting the various results, but rather literally reports the model results in terms of modeled erosion rates etc.. I wonder if the authors can think of presenting a conceptual model to explain and summarize the various model results.

3. We agree with you that the presented P/V - scenarios are only one possible set of outcomes this model can produce. We added a paragraph in the discussion section in reference to a paper from Owen et al. (2010) which we hope helps understanding the general concept we also added another figure which conceptually shows the processes that we think, control the system.

**What was the basis of using 10% and 70% V in the model simulations. I might have missed it.**

4. The basis for using these vegetation-cover values was the idea of creating results for areas that represent end-member states for vegetation-cover and climate. Because we wanted to conduct the same simulation experiments for all areas it was not possible to choose the southernmost area of Nahuelbuta because its surface vegetation cover value is nearly at 100% and we couldn't add to that for the step-change and oscillation-simulations.

**It sound like for a given mean annual P, you need a mean annual V for the sensitivity analysis of the model.**

5. Correct. Observed vegetation/precipitation conditions in Chile do indeed demonstrate a non-linear relationship between mean P and V. This is shown in a figure in the paper. These mean P and V relationships are also present in the Part I (companion) paper. Our use of observed and model supported P and V relationships for the 10 and 70% vegetation cover simulations is also why we chose to set up the model results as 'loosely tuned' to Chile because needed to apply a relationship from somewhere. If we didn't do this, then reviewers would most likely ask what relevance the model selected parameters have to reality. We've modified the text in the methods section to hopefully make this clearer.

The parameter selection was not sufficiently developed. The Manning's roughenss for bare soil is very low for overland flow. What was nV. Kv is very sensitive to n, and keeping n as low as reported in the paper will increase the sensitivity of Kv to low values of V..

6. Thanks for catching this error in the manuscript, the said Mannings' roughness of 0.01 is wrongly reported as Mannings number for fully vegetated conditions. We used 0.01 as Mannings number for bare soil and 0.6 as Mannings number for fully vegetated conditions. fully vegetated conditions were assumed to be the case at V = 1 (100%).

While we realize that 0.01 is a number at the low-end of the spectrum of Mannings numbers, we choose this value because it lies between values reported by Chow (1958) for concrete channels and very hard soils, which we found a feasible assumption for the only unvegetated, granitic bedrock dominated area of Pan de Azucar. We changed table 1 to report the complete and correct values.

The interplay between vegetation and precipitation on erosion rates and landscape evolution is the relatively novel aspect of this paper. I don't think this was discussed in earlier papers, especially by separating V and P scenarios and then combining them based on the dependence between V and P. The one comment I have on the reconstruction of P and V is that, the paper relates P to V as far as I can tell (Fig. 5). I have a hard time rationalizing this as clearly V responds to P, rather than the opposite.

7. This is a good point. We decided that we wanted to tackle the problem from the side of vegetation-modulation and adjust the climatic parameters to certain imposed changes of vegetation cover. We had initial results from a dynamic vegetation model (LPJGUESS) which suggested a 10% change of vegetation cover in those areas since the last glacial maximum, which was the basis for using a 10% transient change in vegetation cover. However, please note that our selection of P and V values for simulations is based on present day observed relationship (empirical) from Chile. Thus,

we are implementing what is currently the case in Chile, rather than assuming some functional relationship between the two.

I wonder if the results, especially for the last case where a complex response as observed, would be any different if V was predicted from P, and P oscillated using a sin function.

8. Again, we like to point out that the paper presents only a suit of possible results. As referred to in the paper, the outcome of the transient simulations is a complex interplay between the initial vegetation cover/climate state of a model-domain and the imposed change in climate and vegetation cover. If we would have chosen to start with fixed values of P and then chose the according vegetation cover, it would boil down to: 1. the magnitude of changes in P that were chosen for the simulations and 2. the transient conditions would resemble a symmetric distribution of dP for step-changes and oscillations and a non-symmetric distribution of dV which would of course influence the results in a way that it would be another possible set of results.

The paper is long. The authors report details about model results as V and P varied. Details such as the rate of erosion etc can be omitted as these are apparent in the plots and in a theoretical study that does not claim to represent a certain region the exact rates of erosion would probably not interest the reader. I suggest focusing more on the conceptual findings of the paper.

9. We have tried to shorten the paper as much as possible. While we agree that the absolute values of rates of erosion may not be applicable to comparison to field-data, due to a possible offset in steady-state rock uplift rates, we think that there is value in reporting these rates and parameters in the text, because it makes the paper easier to read, for readers who may be only interested in a selected few of the results section and the text helps understanding the figures. Furthermore, if we did not report values for change, then reviewers would ask us to quantify statements such as "large increase", or "small change". So, we've tried to streamline the text as much as possible, but have left some values in to hopefully keep readers with diverse expectations for writing style happy.

To that end, however the model cases considered seems to be limited. The paper describes very interesting responses of erosion and landscape evolution driven by changes in vegetation and precipitation, separately and in combination. However given this theory it would be important to discuss when these cases occur.. For example, given the complex response presented in the last scenario, I wonder how plausible is the modeled complex response, are there any observational evidence on this in super arid regions?

10. As the reviewer notes, a key aim of the study is documenting the effects of precipitation and vegetation change on denudation. As these two factors (P and V) always change together we present an individual analysis of each so their relative contributions can quantified. There are no observations available to test the step change or sinusoidal change of P or V independently, but our modeling provides a means to understand the relative contributions of each. The manuscript currently emphasizes the reasoning for exploring P and V changes with a systematic increase in model complexity. For the coupled experiments presented at the end of the paper, the combined effects of P and V changes are shown. The timescale of the changes investigated is for 100 kyr variations in P and V. To the best of our knowledge, there are as of

yet no observations available with sufficient temporal resolution to compare model results to. Type of data that would be needed are a series of terrace or lake deposits that are well dated and contain paleo denudation rate information. While temporal variations in denudation rates are document in different places around the world (e.g. Marshall et al. 2015; Schaller et al. 2001; Schaller et al. 2016) few of these document vegetation, precipitation, and denudation rates over the timescales investigated here.

To address this comment: We have added text similar to the above and the previous references to section 5.5. We also highlight that this study provides a testable hypothesis for future observational studies to consider testing.

Important limitation to realize is that vegetation is spatially uniform, and it does not have seasonality in response to radiation and weather. Rainfall is also seems to be steady state although it was not mentioned in the paper. Such variability, if included, can effectively lead to crossing of erosional thresholds with certain frequency. The reason I'm bringing this up is that the model shows a muted response to Vegetation oscillations for V=70%. While this is high veg cover and the variability may be less in the erosional response, however absence of seasonal veg dynamics and stochastic rainfall and vegetation loss due to scour might play a strong role in stabilizing the land. If these above mentioned processes were used, with steeper equilibrium slopes under V=70%, the model would have responded in some episodic fashion.

11. This is an excellent point - and we partially agree. To address how seasonality influences denudation rates would be a separate study on it's own. You are right that we decided to use a steady-state rainfall approach without internal seasonality. We chose this as our starting point for looking at P and V interactions, but clearly future work on seasonality and stochastic precipitation effects need to be considered as well.

To address this comment: We will highlight the effects of seasonality and stochastic precipitation effects more in the paper as limitation of our approach and caveats that need addressing in the future. We agree that an event-driven climate within the model would introduce variability in the results but due to a lack of good data-proxies in this region to derive a high-resolution timeseries of climatic events would not lead to a gain in information from the paper. We acknowledge that your idea is to have a complete theoretical conceptual model paper and then this approach would be feasible but we would stay with our idea of using the Earthshape areas as proxys for climate and vegetation data within the model.

Line 69: Yetemen et al., 2015a, b are also exceptions to this statement as these papers presented daily water balance, runoff, distributed energy balance and evapotranspiration and transient vegetation growth.

**12. Thanks for bringing that to our attention, we will modify the manuscript accordingly**

Lines 160-162, a citation would be great here on the use of 100 kyr cycles.

**13. We added a citation to build up on the 100 kyr cycle concept.**

Lines 163 – 169: Were the results vegetation dynamics model results with cyclic climate not used.? Perhaps the actual data used to construct Fig 5 may be shown. I also noticed that in Fig 5 while Veg cover oscillates following what looks like a sinusoidal curve, the Precip data oscillate differently. Does lines 167-168 explain the reasons for this, which I was not sure if I understood correctly. For each 10% in veg cover you change the % in precip. But was this assumption

concluded from Fig1, which would give us % V change for %P change within the ranges of P used in the model ? Given precip drives vegetation shouldn't logically Precip be following the sin curve rather than Veg, unless there is a better explanation and rational telling the reader why the curves were plotted differently. I could not find where 10% constant vegetation and near zero precip was mentioned. Some more details on the vegetation cover and the erosional history of the region would be good to include. For example are there any studies that quantified the erosion rates in the Holocene in this region.

14. The +/-10% vegetation cover change was determined by preliminary results from a dynamic vegetation model presented in the companion paper (Part I, Werner et al), but the direct output of the model results were not used in this paper because we wanted to reduce complexity. Actually Fig. 5 shows the actual data from Modis/Worldclim. We will highlight this more in the paper. Again we decided to let P follow V because we wanted the paper to focus on the effect of vegetation change and not changes in precipitation with changes in vegetation as a by-product. We also added citations which we think help with interpreting the complex erosion-rate results of the paper.

Equation 1: please change slope to curvature in the hillslope diffusion term. In the text below the equation kdS is correctly defined as flux but the equation does not use flux it uses the divergence of this flux that leads to change in elevation [L/T] and therefore curvature instead of slope is used. Also if you use curvature in this form the sign in front of the hillslope diffusion component of elevation change should be positive as slope would need another negative sign when represented by elevation change..

15. Thanks for bringing this to our attention, we will change the manuscript accordingly.

Equation 5, Is the E threshold used in the model experiments?

16. Yes, the erosional threshold was used. We added text to the manuscript to make this clearer

Equation 8 was given incomplete. It's missing the shear stress partitioning part that gives the drop of effective shear stress with V, which is correctly plotted in Fig 6.

17. Thanks again for noticing this. We changed the manuscript accordingly.

See equations 10 and 11 in the cited paper in this section. In this model how is rainfall incorporated.. The rate and duration of rainfall would influence the selection of the threshold to make the model results realistic.

18. As mentioned, we used a steady-state mean annual precipitation approach with no stochastic distribution. We will add some text to the method section to make this clearer.

Line 220: please tell what this steepness index was and provide citation. Did you extract the channel network to calculate this?

19. Yes – we assumed a threshold area for channelized conditions and extracted the channel network from our topographies with this. The steepness index was the normalized channel steepness index as for example reported in Wobus et al. 2006

**Line 221: Fig 2 captions says 90 m DEM used.**

**20. Thanks for noticing this. We will change the figure caption.**

Fig 7 d, e, f.. Lines 259.. In calculating channel steepness how were channels extracted? Here the authors make comparison between model predictions and the actual landscape.. Earlier the reader was informed that the model was not specifically calibrated for this region and the purpose of the study is to investigate the model sensitivity to V and P and compare the general trends between observations and models. Here the model is compared directly with actual data values from DEMs and the authors point out under and over estimations of the model.. Given that the model was not calibrated these statements undermine the strength of the paper. This model have enough parameters to calibrate with which the Fig 7 d,e,f can look a lot better.. This brings a few questions on model parameter selection. For example how was erodibility selected? I presume m and n exponents are also constant. Manning's coefficient for full vegetation is very low and unrealistic. This value is more smooth concrete channel value. Can the authors elaborate if any calibration at all was attempted? If the authors want to stress on the general patterns predicted by the model rather than a poor direct comparison between model and observations, they can report these figures in a non-dimensional form so that the amplitude of responses are compared with respect to 1 in both model and data. A simple way to non-dimensionalize would be to dive all the values with the mean (slope, relief etc.).

21. We will try to make the parameter selection clearer in the method-section of the paper. While the model was not parameterized to exactly fit the data from the Earthshape focus sites, we used the general behaviour of topographic metrics between the sites as a proxy for how we wanted the model to behave. Parameters where then chosen from published studies for the granitic lithology dominant in the focus sites and combined with vegetation cover data and climate data extracted from the Earthshape sites.

Fig 8 and 9 results make sense..

Fig 13. This figure shows that for the case of denser vegetation cover (V=0.7 or 70%) and larger P erosion increase with P in the similar way when V=10% (and smaller P), but as P gets smaller in the drier phase of the oscillation erosion does not drop as much as the less dense (and drier) simulation. Why the model gives this asymmetric response in E for given P oscillations (for drier and wetter P) for V=0.7 needs to be explained by the authors, as this is a very interesting result. Also why does the simulation with V=0.7 can double its erosion similar to V=0.1.? The mean erosion is higher than the case with V=10%. 10 cycles were plotted, given a total model year of 1M. I wonder why only the negative changes in P was dampened by vegetation in the denser V case but not the wetter cycles? This probably has to do with the way slopes adjusts under the two climate regimes. In both simulations you have a steady-state landscape as initial condition, and when P grows, is the erosion threshold surpassed in all locations resulting in a very similar erosion magnitude?. How long would the landscape need to attain a dynamic equilibrium under cyclic climate?

22. Thanks for thinking about these results and how to better explain them to readers. We will add some text to make this clearer.

Fig 15.V and P > or < V and P is not very informative.

23. We will make some adjustment to the figure in a more informative way.

**Response to SC2**

Responses in blue

This manuscript has a lot of value in terms of exploring the implications of some of our current best guesses with respect to how vegetation cover and precipitation modulate erosion rates at the Earth surface and influence topographic evolution over long timescales. While the investigation of the individual influences of precipitation and vegetation cover provide insights into the role of each, the dynamic behavior that arises through the combined modeling of both comprises a testable set of results, which could lead the way to improved equations in the case of clear mismatches. People in our community with a focus on details of individual processes and individual types of vegetation cover may find the generalization necessary for such models running over such long timespans to be an oversimplification, but I agree with the approach of first testing whether or not this simple approach can explain first-order observations with respect to erosion rates and landscape morphology.

Along these lines, I think the authors could expand their discussion somewhat to better emphasize how those of us from the field-data side might help to test these results. Testing the results of the model with respect to the morphology of the Andes is useful, but also has some broad limitations. For example, it's difficult to know if the topography that we see today is fully adjusted to forcing conditions such that it reflects the predicted influence of precipitation and vegetation cover, or if there could be a persisting transient response to tectonic activity.

1. Thank you for this productive and thoughtful comment. We agree, that it is hard to quantify if the modern-day topographies which are present in the Andes are in steady-state with the current forcings. We tried to rule out any tectonic transient leftover by picking basins which showed no remnant knickpoints in their river profiles. We agree that ruling out any other transient remnants of changes in vegetation and precipitation is hard, if not impossible to achieve, without additional field data. However we would like to point to publications from Mutz et al. 2018 and especially Schaller et al. 2018, which shows that the climate gradient the Coastal Cordillera has stayed relatively constant for long time periods and therefore we would argue that, if a transient signal of climate change is remaining in these basins, the general trend regarding the N-S gradient should still be comparable.

To address the reviewers comments we will add additional text in section 5.6

Calibrating the model to the conditions in the Andes could be problematic if the landscape is currently in a transient adjustment state. I wouldn't suggest that this issue rules out the possibility of comparing model predictions with topography, but it does point to a reason why there might be mismatches.

2. We fully agree with the reviewer with that the reason for mismatches between model setup and topographies could be due to not fully transient adjustments. As we mentioned in the paper, also a mismatch between the proposed uplift rates the real uplift specific could playing and rates for the areas be into this. Still we also think that its useful to try to reproduce a topographic trend seen in those catchments with our model setup to test for a correct sensitivity of precipitation and vegetation.

Another possibility for comparison would be to consider datasets that have recorded temporal variations in erosion rates in response to changes in precipitation and vegetation cover, particularly over the precipitation and vegetation cover ranges that the authors suggest drive the biggest changes. Field data that support the model results (particularly Figure 17) would be a strong argument in favor of the equations used. Unfortunately, those datasets are somewhat rare; the only ones I am aware of that report both vegetation cover and precipitation changes include Marshall et al. (2015, Science Advances) and Garcin et al., 2017 (EPSL), although Marshall et al. was mostly focused on frost-cracking, which would be less relevant for the model presented here. While I'm a co-author of the second and don't want to insist that the authors consider that study, I think it could be valuable, as we found quite strong variations in 10Be derived erosion rates reflecting the onset and persisting influence of the African Humid Period. An in-depth comparison of your model results to those data would clearly be out of the scope of the manuscript, but a qualitative comparison could be useful.

3. We agree that field data which supports our model results is unfortunately hard to to find. Owen et al. (2010) however discussed CRN-derived soil-production rates in the dry regions of Chile. If the assumption of a constant soil-column thickness holds true these soil production rates can be translated into denudation rates. In this paper in Fig. 12, the authors find that soil production rates for areas with a MAP between 2 mm and 10 mm show a huge variation with lowest values of 0.1 - 0.2 m/Ma which would translate into 0.0001 - 0.0002 mm/yr. They report these values to be situated into either a abiotic zone for 2mm MAP or a transition zone for 10 mm MAP. The maximum denudation rates they report in their dataset increase from 2m/Ma to 4m/Ma which shows that while the mean annual precipitation might increase by a factor 10, the denudation rates show no significant increase, moreover the minimum denudation rates of nearly zero despite an increase in precipitation. We argue that this dataset supports our model results by showing that an increase in precipitation from an abiotic regime to a "transitional" regime is not necessarily associated with increasing erosion rates but zero erosion rates could be plausible for this transitional regime.

To address the reviewer's comment we will add the mentioned citation and modify the discussion section to incorporate the above reasoning.

One aspect of the modeling results that surprised me is that for the 10% vegetation cover case, increasing precipitation leads to a decrease in erosion rates in the model. I find this somewhat counter-intuitive, and contrasts in some ways with what we see in East Africa, where the onset of wetter conditions leads to a brief spike in erosion rates, however the rates rapidly decreased once denser vegetation cover (trees, compared to a mix of grasses and trees earlier) established itself. So why the difference with your results? I'm guessing there's not much of a time lag in the model between increased precipitation and increased vegetation cover... if there were, perhaps the model would show a response more similar to what we see in our data. I'm unsure whether short short-term responses are important for landscape evolution models running on million-year timespans. But even if it's brief, a spike in erosion rates could have a reasonably good change of being preserved in stratigraphy, so there may be a good chance of recovering such responses with field data.

4. The reviewer is right in assuming that our model does not incorporate a lag-time between vegetation growth and changes in climate. Because of the different reactions of specific plant functional types to climate changes and general shifts in ecosystem inventories it is hard to quantify lag times of vegetation cover to shifts in precipitation. There are however ecological studies that tried to understand the lagtime of specific systems and plant types to changes in precipitation (e.g Fensham et al. 2005) and in general it can be concluded that these changes happen on timescales of 10 - 100 years. While we acknowledge that these adjustments are highly sensitive to the initial conditions of the systems and

that other disturbances like fire frequency etc. also play a role in the time which the ecological systems need to adjust, we argue that the lagtimes will still be in the same order of magnitude than our model timesteps (dt = 100yrs) and will therefore not change the outcome of the results. However we would fully agree that these short lagtimes could cause for a short burst of high erosion rates after onset of precipitation until the vegetation is adjusted, we only can't see it in the model without switching to smaller timesteps, which would be computationally efficient for the questions we hope to answer.

However, because we agree that the manuscript could be improved by linking our results to field data and other 'less theoretical' studies, we added a new subsection: "5.5 Potential Observational Approaches to Test Model Predictions"

Along the same lines, I wonder whether or not it would be possible to record (with field data) the high-frequency variations that the model shows during the declining-precipitation phase (again for the 10% veg cover case). Do you think we could we resolve a sudden, brief decrease in erosion rates with 10Be data? Two reasons why it could be tricky is that the thickness of any stratigraphic layer would be limited, and also at lower erosion rates, the erosion rates are integrated over longer timescales.

5. The reviewer brings up an interesting point about the applicability of field-methods to resolve the nearly-zero erosion intervals that our model predicts. We believe that in general it would be possible to resolve these phases with cosmogenic nuclides in a perfect case with a complete set of deposited undisturbed sediment-layers. However even for erosion rates of 0.1mm/yr, the integration time for 10Be would be around 6000yrs, which is longer than the zero-erosion interval our model predicts, so to exactly resolve this case will prove difficult.

This point may be worth discussing. Some additional minor points:

Section 4.1, lines 248, 255, 266: It would be helpful to use the term "spatially variable" in this section rather than "variable" alone, to make it clear that you are not focusing on temporal transients.

Thank you for this input. We believe that using the term "spatially variable" could mislead readers to think that we use a spatially variable vegetation cover within each model domain. However we agree that it could be more clear, so to address this we will change the text to make this point more clear.

**Why isn't section 5.4 in the results section (section 4)?**

We wanted to set-up the paper so that the coupled simulations that we ran could be thought of as a addition which helps to understand the previously set-up simulations with isolated transients forcings of either vegetation cover or mean annual precipitation. Our presentation of the coupled results in section 5.4 also provides a way of integrating the results, which if we removed it from the discussion section we would likely be asked to address by other reviewers (some of them already asked for more integration). Perhaps this is a difference in writing style, but we prefer to keep the fully coupled simulations in the discussion section to help manuscript integration.

Figure 16 is hard to interpret without Figure 17; it would be easier if the forcing (change in precip) were shown in Figure 16 or at least explained in the caption.

Thanks for this suggestions, we will add a more thorough explanation about the forcings in the figure caption for figure 15, for this figure is the first one that shows the results of the coupled simulations.

**Are changes in vegetation cover modeled or prescribed?**

I'm embarrassed that I was reading too quickly to discern that detail, but in any case, it would be helpful to have that information more prominent.

Changes in vegetation cover were prescribed and set to 10%. We added text in section 3.2 to make this clearer.

[revised manuscript text omitted]
 2013. Those approaches mostly differ on the degree of underlying details which were used to parameterize the model and the claim of reproducing certain aspects of landscapes on a temporal and spatial scale which heavily depends on the used approach (for details about the different approaches, see Dietrich et al., 2013). For this study we have chosen the approach of essential realism, which acknowledges a system-inherent indeterminacy in the evolving topography but focuses on predicting the first-order trends within a system and the differences between landscapes, based on different external conditions, incorporated in the model (Howard, 1997).

While we do not claim to reproduce the topographic metrics of the four different focus areas in Chile on a realistic level, our approach determines the general first-order effects of millennial timescale changes in precipitation and vegetation cover that can impact topography. Superimposed on the effects documented in this study would be the effects of seasonal changes in precipitation and vegetation cover, subcatchment variations in vegetation cover, transport limited fluvial and vegetation interactions, stochastic variations in precipitation in different climate zones. Consider of the previous, more detailed, aspects of precipitation-vegetation interactions on erosion could be independent studies of their own and can not be covered in a single study. Thus, the modeling approach and results of this study should be considered as documenting the longer (millennial) timescale climate and vegetation forcings on fluvial and surface processes.

**3. Methods**

**3.1. Model Description and governing equations**

[revised manuscript text omitted]

---

## Author Response (AR2)

**1 Letter to the Editor**

We like to thank Rebecca Hodge again for the thoughtful and productive handling of the manuscript so far. We hope that these revisions helped in making the manuscript better and more valuable to the scientific community. We addressed all of Rebecca Hodge's comments and thought of all of them to be justified and agreed with all her suggestions. As discussed with Rebecca Hodge, we here just present a tracked-changes version of our manuscript and no point-by-point answer to her comments.

Manuel Schmid and Todd Ehlers (corresponding author) on behalf of all the co-authors.

[revised manuscript text omitted]